



# An observation operator for geostationary lightning imager data assimilation in the French storm-scale numerical weather prediction system AROME

Pauline Combarnous[1,2], Felix Erdmann[3], Olivier Caumont[1,4], Éric Defer[2], and Maud Martet[1]

[1]CNRM, Université de Toulouse, Météo-France, CNRS, Toulouse, France
[2]LAERO, Université de Toulouse, UT3, CNRS, IRD, Toulouse, France
[3]Royal Meteorological Institute of Belgium, Brussels, Belgium
[4]Météo-France, Direction des opérations pour la prévision, Toulouse, France

**Correspondence:** Pauline Combarnous (pauline.combarnous@meteo.fr)

**Abstract.** The Lightning Imager (LI) onboard the Meteosat Third Generation (MTG) satellites will provide total lightning observations continuously over Europe with a spatial resolution of a few kilometers. The objective of this study is to prepare the assimilation of the flash extent accumulation (FEA) measured by LI in the French storm-scale regional AROME NWP system within a new EnVar assimilation scheme, by developing a lightning observation operator. This study relies on pseudo
LI FEA observations as LI on MTG is still to be launched in the end of 2022, meaning actual observations will be available by 2023. Since neither flashes nor the electric field are predicted by the AROME NWP system, the observation operator relies on proxy variables to link the flash observations to the prognostic variables of the NWP system. A total of 8 storm parameters were selected from a literature review to be used as proxies. Two different proxy types emerged from this literature review: microphysical and dynamical proxies. The proxies are calculated from 1h AROME forecasts from the assimilation
cycle. Machine learning regression models are used to relate observed FEA and the simulated proxies. The training of the observation operator is performed on a dataset of 44 days and 3 additional days are used for the validation. The data are processed as a climatology over the whole domain (i.e. France) and time period. The performances of each proxy are evaluated by computing Fraction Skill Scores (FSS) between observed FEA and proxy-based FEA. The present study suggests that microphysical proxies seem to be more suited than the dynamical ones to model satellite lightning observations with the
AROME NWP system. The performances of multivariate regression models are also evaluated by combining several proxies after a feature selection based on a principal component analysis and a proxy correlation study but no proxies combination yielded better results than microphysical proxies alone. Results are also compared to those obtained with another lightning calibration, i.e., a relationship between lightning and proxies, from the literature but the simulated FEA amplitudes were systematically lower than the observed ones. Finally, different accumulation periods of the FEA had little influence, i.e., similar
FSS, on the observation operator's ability to reproduce the observed FEA.



# 1 Introduction

Thunderstorms that produce phenomena such as lightning, flash flooding and hail are extreme, dangerous and destructive events. A better short-term prediction of those convective events could help to prevent some damages and warn the population with sufficient lead time. Nevertheless, in spite of the continuous improvement in numerical weather prediction (NWP) systems, thunderstorms remain hard to predict with high accuracy. This difficulty partly results from a lack of storm-related observations to describe the initial state of the atmosphere, especially over regions like oceans, mountains and countries without ground-based radar networks.

Total lightning, i.e. cloud-to-ground (CG) and intra-cloud (IC) lightning, is a good indicator to pinpoint thunderstorms and evaluate their severity. According to Carey et al. (2005), flashes tend to be initiated and are more frequent near strong updrafts while smaller in size than flashes in stratiform regions of the cloud (Weiss et al., 2012). It has also been shown that a fast increase in the lightning activity, i.e. "lightning jump", is a precursor of thunderstorm intensification (Schultz et al., 2009, 2011).

Because of the link between lightning activity and thunderstorm characteristics, lightning observations represent a potentially interesting source of information to initialise NWP systems by adding lightning data assimilation (LDA). Several studies have already demonstrated the potential of LDA with different assimilation approaches at convection-permitting resolution. Nudging methods have been employed by Marchand and Fuelberg (2014) and Fierro et al. (2012) and methods using an ensemble Kalman filter have been developed by Mansell (2014) and Allen et al. (2016). Variational approaches were investigated in more recent studies by Fierro et al. (2016, 2019); Zhang et al. (2017); Hu et al. (2020); Xiao et al. (2021). A general improvement was observed in accumulated precipitation for short term forecasts ($\leq$3h) when lightning data is assimilated.

The lightning data used in most of those studies (e.g. Marchand and Fuelberg, 2014; Fierro et al., 2012, 2016, 2019; Hu et al., 2020) are either data from the Geostationary Lightning Mapper (GLM) on the Geostationary Operational Environmental Satellite-R (GOES-R) or data mimicking GLM products when the study was prior to its launch. Satellite lightning data present numerous advantages including their large spatial coverage, providing observations where ground-based radar data are scarce or nonexistent and making them well-suited for convective-scale data assimilation in limited-area NWP systems. By 2023, lightning observations from the Lightning Imager (LI) onboard the Meteosat Third Generation satellite (MTG) will be available over Europe, Africa, the Atlantic Ocean and a part of South America. The LI will be able to detect both IC and CG lightning, providing total lightning information, day and night. One of the products that will be provided by the LI is a flash count per pixel accumulated over time and hereafter referred to as flash extent accumulation (FEA)[1].

In modern data assimilation systems like variational or ensemble based systems, an observation operator is required to establish a link between observations and the NWP system background. For lightning observations, developing an observation operator is not trivial since most operational NWP systems do not include an electrification and lightning scheme. Indeed, the equations to relate the NWP system prognostic variables and the microphysics to produce an electric field are complex, nonlinear and computationally expensive (e.g. Barthe and Pinty, 2007; Mansell et al., 2002). In consequence, the observation

---

[1]Note that the FEA is referred to as flash extent density (FED) in former studies but after a reflection on the spatially intensive nature of this value, the terminology FEA was eventually adopted


operator developed in this study is based on empirical relationships between the lightning observations and a set of proxies derived from the NWP system variables, in a similar approach as used in the assimilation papers mentioned above.

The objective of the present study is to prepare the assimilation of lightning satellite data in storm-scale NWP systems by designing a suitable observation operator in the scope of development of a new EnVar assimilation system for the French regional NWP system AROME-France. The lightning data employed here are generated from ground-based lightning data from the Météorage network (Erdmann et al., 2022) to mimic the future MTG-LI data. This paper is organized as follows. In sections 2.1 and 2.2 the lightning generator to mimic the MTG-LI data as well as the NWP system configuration are described. The list of the tested proxies is detailed in section 2.3, along with the way they are retrieved from the prognostic variables of AROME-France. Section 2.4 depicts the method to link the lightning pseudo-observations to the selected proxies. The results for 10 min FEA are presented in section 3. In section 4, both lightning threats introduced by McCaul et al. (2009) are also investigated and the McCaul lightning calibration is reproduced to be compared with the results of the method described here. The sensitivity of the relationships established in section 3 to the FEA accumulation period is then studied in section 5. Finally, section 6 discusses and summarizes the main conclusions of the paper.

## 2 Data and methodology

In the context of the present study, 1 h forecasts from the AROME-France assimilation cycle for 47 stormy days in 2018, described in Table 1, were used to compute a set of 8 storm variables to be tested as proxies for MTG-LI observations. Those storm variables are hereafter directly referred to as "the proxies". Synthetic MTG-LI observations were generated for the same 47 days as targets to train observation operators first from individual proxies and then from combinations of proxies based on their individual performances and their inter-correlations built on machine learning (ML) regression algorithms. In the following, a brief description of the lightning generator and the NWP system configuration is provided, as well as a detailed description of each proxy and how they are retrieved from the AROME-France forecasts. Eventually, the pre-processing applied to the data, the regression models and the verification metrics are described in the last subsection.

### 2.1 MTG-LI synthetic observations

As mentioned in the introduction, the geostationary LI will provide observations by 2023. The geostationary LI is an optical sensor that detects the cloud top illumination due to lightning with a resolution of 7 km at European latitudes and 4.5 km at its nadir. Total lightning activity will be detected both day and night but LI detection efficiency will most likely vary with the time of the day, and possibly with the geographical position of the lightning activity within LI field-of-view. This assumption is based on related studies on the LIS instrument onboard the International Space Station (ISS-LIS) and the GLM (Bateman et al., 2021; Peterson et al., 2017).

Erdmann et al. (2022) developed a method to generate GLM pseudo-observations using lightning data from the National Lightning Detection Network (NLDN), consisting of ground sensors in the contiguous US. As the MTG-LI and GLM instruments are expected to provide similar observation and data structure, and an intercomparison study between the NLDN and
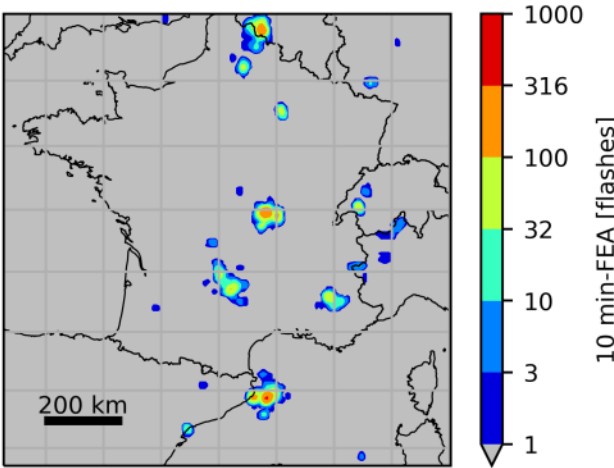

**Figure 1.** Synthetic MTG-LI observation generated in the domain of interest (in gray), accumulated over 10 min between 20:55 and 21:05 UTC on 7 August 2018.

the French Météorage low frequency network performances relative to the ISS-LIS instrument showed good consistency (Erdmann, 2020, Chap. II) the lightning generator can be used to produce synthetic MTG-LI observations using Météorage data as entry.

The simulation of synthetic observations is performed in two steps. First, GLM (or MTG-LI in our case) flash characteristics, such as flash duration or flash extent, are simulated using ML supervised models (Erdmann et al., 2022). The ML models have been trained with flash characteristics of coincident NLDN and GLM flashes from a database of 10 lightning active days in the South-East of the USA, spread over a 6 months period. The second step consists in generating the pseudo FEA gridded observations from the simulated flash characteristics. The method was evaluated by comparing the GLM synthetic observations with operational GLM data. Erdmann et al. (2022) recommend the use of the linear Support Vector Regressor (LinSVR) based generator since it is the one yielding the best results overall. Consequently, all the MTG-LI synthetic observations used in the present paper are simulated with the LinSVR generator. A typical example of a synthetic MTG-LI observation generated over France used in this study is presented in figure 1, for an accumulation period of 10 min between 20:55 and 21:05 UTC on 7 August 2018.

## 2.2 NWP system configuration

The AROME-France NWP system (Seity et al., 2011; Brousseau et al., 2016) resolves deep convection with a horizontal resolution of 1.3 km. It is operational since the end of 2008 with a major update in 2015 concerning its resolution and a reduction of the period of the data assimilation cycle from 3 to 1 h. It is a limited-area NWP system with a geographical domain shown in Figure 2. AROME-France is computed on 90 vertical levels with a maximum altitude of 10 hPa but the levels are mostly concentrated in the troposphere. The model simulates twelve prognostic variables including two components


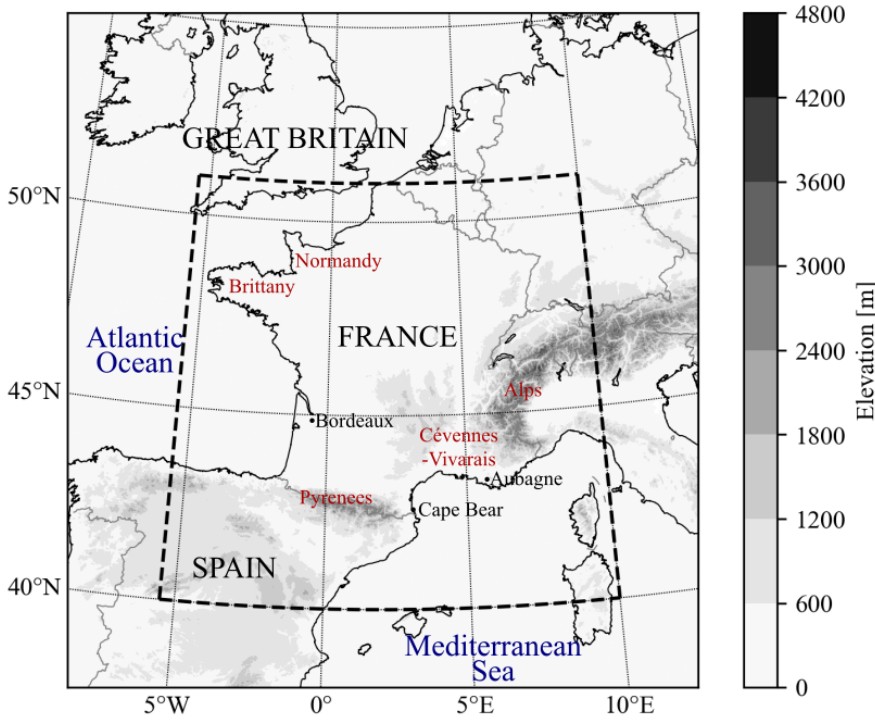

**Figure 2.** The AROME-France domain and topography. The dashed rectangle shows the limits of the sub-domain where the synthetic MTG-LI data are generated. Locations mentioned in the text are also indicated.

of the horizontal wind ($U$ and $V$), the temperature $T$, specific content of water vapour $q_v$ and the surface pressure $p$. The ICE3

microphysical scheme used in AROME-France predicts 5 out of the twelve prognostic variables: specific contents of rain $q_r$, snow $q_s$, graupel $q_g$ (including different types of large rimed crystals like graupel, frozen drops and hail), cloud droplets $q_c$ and ice crystals $q_i$. The proxies are derived from 1 h AROME-France forecasts from the assimilation cycle and represent the state of the atmosphere at a fixed time (not accumulated). The choice of the hourly forecasts is based on the duration of the current assimilation cycle of the NWP system.

**2.3  Description of proxies**

This section focuses on 8 different storm parameters and their link with the FEA. The selected proxies are the following: the ice water path (Petersen et al., 2005), the updraft volume (Deierling and Petersen, 2008), the graupel mass (Deierling et al., 2005), the maximum vertical velocity (Deierling and Petersen, 2008), the rimed particle column (Figueras i Ventura et al., 2019), the lightning potential index (Yair et al., 2010), the upward graupel flux (McCaul et al., 2009), and the vertically

integrated ice water content (McCaul et al., 2009). They all have already demonstrated a strong correlation with either lightning density or lightning flash rate and represent both dynamical and microphysical properties of the cloud. Proxies are considered


"microphysical" if they rely on ice masses and "dynamical" if they depend on vertical wind velocities. The original studies where each of those proxies appears as well as the way they are retrieved from AROME-France background are described below. Note that these proxies are not obtained in the same way as in the original studies, where they were either simulated with different NWP system than AROME-France or measured with instruments with their own specific sensitivity and spatio-temporal resolution. Here, all the proxies are calculated column-wise from AROME-France fields. Hence, the objective of this work is to test their ability to reproduce lightning observations with one proxy value per AROME-France column and results may not be comparable to original studies.

Other proxies were also investigated in the literature. A lightning parameterization based on a 5th power dependence between the flash rate and the cloud top height has been established by Price and Rind (1992) (PR1992 hereafter) and introduced by Wong et al. (2013) in the Weather Research and Forecasting (WRF) model. Nevertheless, several case studies evaluated this relationship and either recommended that other thermodynamical and microphysical variables should be used as a masking filter (Giannaros et al., 2015), or to use the cold cloud depth instead (Yoshida et al., 2009) or found a relationship that slightly differs from PR1992 parameterization (Karagiannidis et al., 2019). As cloud top height alone does not seem to perform well to pinpoint lightning activity, the PR1992 parameterization was not included in this study.

**Ice Water Path**

Petersen et al. (2005) studied the relationship between lightning flash density and the vertically integrated precipitation ice masses (referred to as the Ice Water Path or IWP hereafter) on a global scale using three years of TRMM Lightning Imaging Sensor (LIS) and Precipitation Radar observations. They transformed pixel-level IWP and LIS flash counts to $0.5° \times 0.5°$ grids. Two methods were then used to compare the IWP ($\mathrm{kg\,m^{-2}}$) and the flash density ($\mathrm{fl\,km^{-1}\,day^{-1}}$) statistics: The first method uses IWP and flash densities coincidentally observed over the area of individual $0.5°$ grid elements during each TRMM overpass and the second method uses time-integrated means for those individual $0.5°$ grid squares. For a more detailed description of the methods, see Petersen et al. (2005). The relationship between the IWP and the flash density is established with a linear best fit, with a linear correlation coefficient ranging from 0.97 to 0.99, depending on the regime (land, ocean, coastal) and the method. For the purpose of our study, the IWP is calculated for each column from the altitude of $-10°$C to the roof of AROME-France (versus echo top for Petersen et al., 2005) as follows

$$\mathrm{IWP} = \int_{-10°C}^{\mathrm{roof}} \rho(q_s + q_g)\,dz \qquad (1)$$

where $\rho$ is the local air density in $\mathrm{kg\,m^{-3}}$ and $q_s$ and $q_g$ are the simulated specific contents of snow and graupel, respectively.

**Updraft Volume**

The updraft characteristics, such as updraft volume and the maximum updraft speed, were investigated versus the total lightning activity measured by a Lightning Mapping Array (LMA), by Deierling and Petersen (2008) for a collection of thunderstorms in the High Plains and in Northern Alabama. The updraft volume was retrieved using Doppler radars and computed for vertical





velocities greater than 5 or 10 $\mathrm{m\,s^{-1}}$ between $-5$ °C and $-40$ °C. Their results showed a good correlation (linear correlation coefficient $r = 0.93$) between the updraft volume and total lightning activity. However, the vertical velocities of our AROME simulations do not often reach values higher than 5 $\mathrm{m\,s^{-1}}$. Consequently, in the present study, the updraft volume ($\mathrm{m^3}$) is defined as the sum of grid cell volumes with vertical velocity higher than 1 $\mathrm{m\,s^{-1}}$ for each column from $-5$°C to the roof.

**Graupel Mass**

The graupel mass is one of the most investigated storm parameters as a proxy for lightning activity. Deierling et al. (2005) used polarimetric radar data and total lightning measurements from a 3-D lightning VHF-interferometer system to compare trends in hydrometeor types with total lightning frequency. The precipitable ice (graupel and hail) trend was the most correlated one during the studied storm lifecycle (linear correlation coefficient of 0.73). In our analysis, the graupel mass $m_g$ (kg) is computed for each grid cell and summed over the column between $-5$°C and the roof as follows

$$m_g = \sum_{-5°\mathrm{C}}^{\mathrm{roof}} q_g \cdot \rho \cdot V \tag{2}$$

where $V$ is the volume of the grid cell in $\mathrm{m^3}$, and $\rho$ and $q_g$ are defined above.

**Maximum Vertical Velocity**

The maximum vertical velocity wmax ($\mathrm{m\,s^{-1}}$) was studied as an updraft characteristic alongside the updraft volume by Deierling and Petersen (2008). Their study exhibits a linear correlation coefficient of 0.82 between the time series of mean total lightning per minute and the maximum updraft speed for the 11 storms investigated. The maximum vertical velocity in each column of the AROME-France field is used for the present study.

**Rimed Particle Column**

The rimed particle column (hereafter called RPC, in m) is described by Figueras i Ventura et al. (2019) as the difference between the upper limit of the highest level where rimed particles are predominant and the lower limit of the lowest level where those species are predominant. The data they used are from a lightning measurement campaign that took place in the Alps in 2017 where a LMA was deployed for the occasion. The RPC was retrieved out of a hydrometeor classification from radar data. They noticed an increase in lightning activity from a RPC thickness of 3 km onward and a high CG lightning activity when the RPC was larger than 8 km. For the purpose of our study, the levels where rimed particles are predominant are the levels where the specific content of graupel $q_g$ is higher than each specific content of the other hydrometeor variables. As mentioned above, the graupel specific content gathers several types of large rimed hydrometeor types.

**Lightning Potential Index**

Yair et al. (2010) developed a lightning potential index (LPI, in $\mathrm{J\,kg^{-1}}$) as a parameter to predict lightning. The objective of this index is to use the model output of microphysical parameters in conjunction with the vertical velocity field to parameterize





the potential for charge generation and separation within the charging zone (0°C to −20°C). Yair et al. (2010) compared the simulated LPI from WRF for several Mediterranean flash flood cases with lightning observations from two sources: the Israeli Electrical Company LPATS system and the ZEUS European network. In our study, as well as in that of Yair et al. (2010), the

LPI is calculated for each column as

$$\mathrm{LPI} = \frac{1}{V} \iiint \varepsilon w^2 \, dx \, dy \, dz \tag{3}$$

with $V$ being the volume of the column between 0°C and −20°C in $\mathrm{m}^3$, $w$ the vertical velocity of the wind in $\mathrm{m\,s}^{-1}$ and $\varepsilon$ a dimensionless number defined as

$$\varepsilon = 2 \frac{\sqrt{Q_i Q_l}}{Q_i + Q_l} \tag{4}$$

where $Q_i$ is the ice fractional mixing ratio and $Q_l$ the total liquid water mass mixing ratio defined by

$$Q_i = q_g \left( \frac{\sqrt{q_s q_g}}{q_s + q_g} + \frac{\sqrt{q_i q_g}}{q_i + q_g} \right) \tag{5}$$

$$Q_l = q_r + q_c \tag{6}$$

with the specific contents $q_g$, $q_s$, $q_i$, $q_r$ and $q_c$ described in section 2.2.

**Upward Graupel Flux (F1) and Vertically Integrated Ice Content (F2)**

McCaul et al. (2009) developed a parameterization to forecast lightning threat using two different parameters and a blended approach where those two parameters are weighted to take advantage of the strengths of each. The first parameter, hereafter called F1, is the simulated upward graupel flux in the −15°C layer for each column, in $\mathrm{m\,s}^{-1}$, calculated as

$$\mathrm{F1} = w \cdot q_g \tag{7}$$

The vertically integrated ice content (F2, in $\mathrm{kg\,m}^{-2}$) is the second parameter investigated by McCaul et al. (2009). The quantity

is integrated over the whole column as

$$\mathrm{F2} = \int \rho (q_s + q_g + q_i) \, dz \tag{8}$$

with the same variables as previously introduced. Note that in McCaul et al. (2009)'s study, F1 and F2 are the names of the *functions* linking the parameters to the lightning density, whereas in our study F1 and F2 refers to the parameters themselves.

In McCaul et al. (2009)'s study, both parameters are simulated using WRF, with resolved deep-convection, for seven case

studies from the North Alabama region. The flash rate density is derived from lightning observations measured by the North Alabama LMA by regrouping VHF sources into flashes and mapping them on a 1 km resolution grid. However, it is the flash origin density, which assigns a unit value to the grid cell where a flash initiates, that is mainly used in their study. The calibration of F1 and F2 functions is obtained by extracting and comparing the maximum flash origin density and the maximum simulated parameters from each of these seven simulated cases. The functions F1 and F2 linking the flash origin





density and each parameters are estimated with a linear regression using a reduced major-axis technique and an intercept forced at the origin. The study of two successful WRF simulation cases where the observed and simulated flash densities are compared demonstrated that F1 captures much of the temporal variability of the observed peak lightning flash density whereas F2 reproduced the areal coverage of lightning density better. In consequence, the blended approach is a linear combination of both functions, weighted more heavily toward F1 because they estimated that a small contribution of F2 would be sufficient to provide the desired increase in areal coverage. The blended parameter F3 is then defined as

$$F3 = r_1 \cdot F1 + r_2 \cdot F2 \tag{9}$$

where $r_1 = 0.95$ and $r_2 = 0.05$. This blended parameterization is tested for our dataset, using the same calibration technique, and the results are presented in section 4. Both of the parameters are also studied herein individually and treated with the same method as the others proxies described in section 2.4.

## 2.4 Methodology

The aim of our study is to establish a certain relationship between the proxies and the FEA observations. Therefore, we fit linear and non-linear ML regression models to observations and simulations from 44 days of 2018, listed in Table 1, to yield this relationship. Those days are referred to as the training set and were selected either because of their high lightning activity in terms of strokes and pulses detected by Météorage or because severe events relative to thunderstorms such as flooding or wind gusts have taken place. They were also chosen to represent the annual distribution of thunderstorms, with a larger number of days selected on summer months. Then, 3 additional days (7, 8 and 9 August) constitute the validation set to evaluate the relationships established during the training process. One day (26 May) is used for a sensitivity study to the accumulation period (section 5). In the context of this sensitivity study, FEA with accumulation periods of 5, 10, 15, 20, 30 and 60 minutes were investigated. However, it is the 10 min FEA that was used to compare the proxies' performances (following section), centered at the corresponding time of the AROME-France analysis. The variability of the training dataset is studied in Figure 3, showing the domain-wide FEA maximum for each 10 min period of each day of the dataset. The majority of lightning activity takes place between May and October and the diurnal cycle peaks during afternoon and evening time, between 12:30 and 23:00 UTC, reaching FEA values higher than 500 flashes per 7 km by 7 km pixel in 10 min. To compare the representativeness of the training and validation datasets, histograms of all the 10 min FEA values greater than 0 for each 7 km by 7 km pixel within the domain for each 10 min period from both of these datasets are plotted in Figure 4. The null FEA values, accounting for 98.71% and 97.15% of the total amount of 10 min FEA data for the training and the validation datasets, respectively, are discarded. Both histograms are normalized to their maximum binned value, indicated in Figure 4's caption. The validation dataset presents a higher proportion of 10 min FEA greater than 10 flashes than the training dataset, likely due to the selection of the validation dataset as summertime thunderstorms.

First, the proxy data are projected on the FEA grid by selecting the closest value to the FEA pixel center. One cannot expect the NWP system to simulate individual convective cells in the exact time and location as the observed storm because of typical time and space displacements of convection in the system by more than the FEA time and space resolutions. In consequence, a





**Table 1.** List of the days of 2018 used in the present study.

| Dataset | Month | Day of month |
|---|---|---|
| Training | January | 7, 8 |
| | February | 1, 2 |
| | March | 10, 30 |
| | April | 28, 29 |
| | May | 7, 8, 22, 25, 27, 28 |
| | June | 3, 5, 9, 30 |
| | July | 1, 3, 4, 15, 16, 20 |
| | August | 12, 14, 22, 28 |
| | September | 5, 6, 12 |
| | October | 6, 7, 8, 9, 10, 15, 29 |
| | November | 9, 20, 21, 22 |
| | December | 3, 19 |
| Validation | August | 7, 8, 9 |
| Sensitivity study | May | 26 |

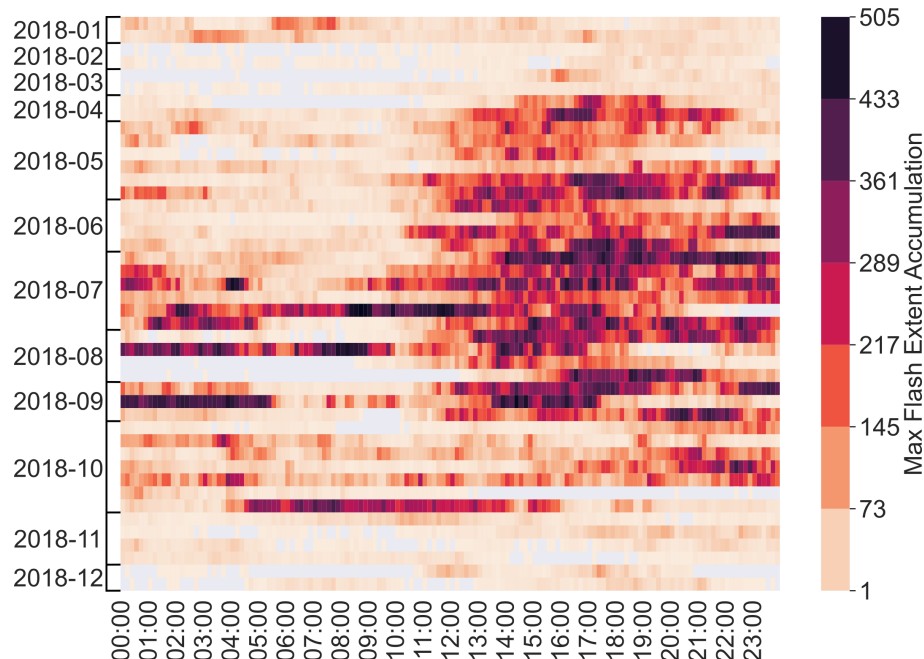

**Figure 3.** Domain-wide maximum FEA for each 10 min of each day of the training dataset. The FEA is calculated on 7 km per 7 km pixels.





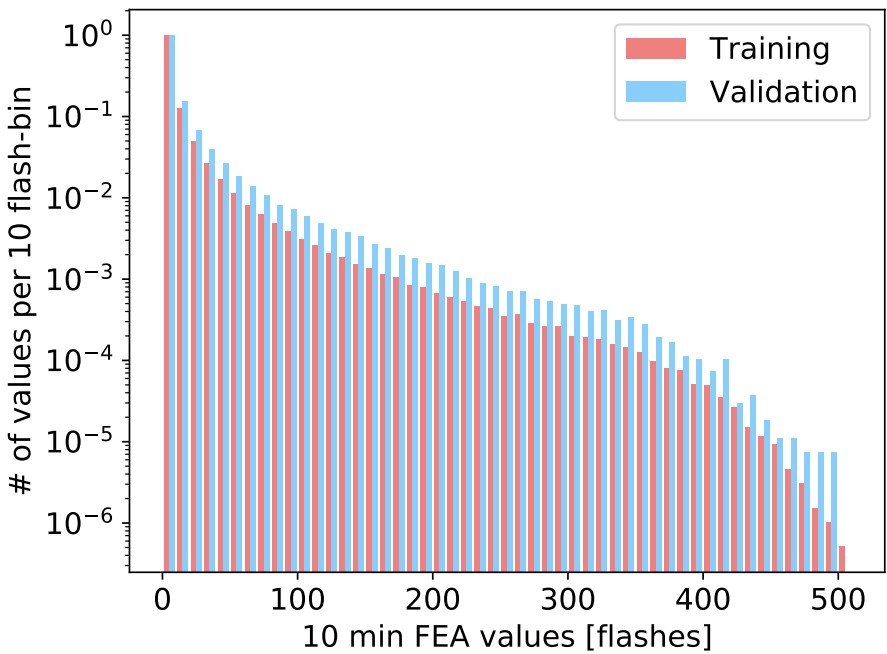

**Figure 4.** Histograms of all 10 min FEA values greater than 0 for each 7 km by 7 km pixel for the training dataset and the validation dataset, both normalized to their maximum binned value, i.e. 1,955,622 for the training and 270,310 for the validation.

pixel-to-pixel comparison is not performed and the data are stacked and rank-ordered over the whole time period and domain (i.e. dashed rectangle in Figure 2) so that they are treated as a global sorted distribution without time and space dependence. Then, all data points with proxy values equal to zero are disregarded. This classification allows to identify a proxy threshold $X_{th}$ corresponding to the first non-zero FEA value, that can be interpreted as the minimum required amount of the proxy quantity to observe lightning.

Finally, the data are normalized to the [0,1] range to perform the regressions. All the results presented in this paper are re-scaled to physical units.

To relate each proxy individually to the FEA, five regression models were tested: an ordinary least squares linear regression (LinReg), a cubic polynomial regression (PolyReg3), a linear support vector regressor (LinSVR), a multi-layer perceptron (MLP) with 20 hidden layers and a random forest regressor with 20 decision trees (RF20). The depth of the trees was unbounded. All the regression models were implemented in Python using the Scikit-Learn package. For the linear regressions (LinReg and LinSVR), the established function $f(\text{proxy})$ is piecewise-defined so that the regression is performed only with the non-zero points

$$f(\text{proxy}) = \begin{cases} 0 & \text{if proxy} < X_{th} \\ a \cdot \text{proxy} + b & \text{if proxy} \geqslant X_{th} \end{cases} \qquad (10)$$




where $a$ and $b$ are the coefficients of the linear regression curve. For the PolyReg3, the MLP and the RF20, the regression is performed on all the data. As the RF20 regression algorithm is a model that cannot extrapolate, the non-zero proxy values that correspond to a null FEA need to be kept to be learned by the model.

After the fit or training, the skill of each model is assessed with the coefficient of determination $R^2$, defined as

$$R^2 = 1 - \frac{\sum_{i=1}^{n}(y_i - \hat{y}_i)^2}{\sum_{i=1}^{n}(y_i - \bar{y})^2} \qquad (11)$$

for an ensemble of $n$ samples where $\hat{y}_i$ is the predicted value of the $i$-th sample, $y_i$ is the corresponding observed value and $\bar{y}$ is the mean of all observation values. Since the $R^2$ score is sometimes very close to 1 in our study, it is calculated as $1 - R^2$ in the plots and Table for the sake of easy monitoring, but the text discusses the $R^2$ score. This score is also computed on the 260 validation set in order to measure the model's predicting ability on an independent dataset.

The Fraction Skill Score (FSS; Roberts and Lean, 2008) is also computed using the fast calculation method introduced by Faggian et al. (2015) to evaluate the displacement error of the modelled FEA values compared to the observed ones for the validation set. The FSS is a neighbourhood verification score where the spatial distribution of events is treated probabilistically which is particularly valuable for data with a small horizontal grid spacing. For fractions $p_o$ and $p_f$ of observed and forecast 265 events, defined as the ratio of the number of pixels with values higher than a fixed threshold and the total number of pixels in a defined neighbourhood size (behaving as a sliding window), the FSS can be written as

$$\text{FSS} = 1 - \frac{\frac{1}{N}\sum_{i=1}^{N}(p_f - p_o)^2}{\frac{1}{N}\sum_{i=1}^{N}p_f^2 + \frac{1}{N}\sum_{i=1}^{N}p_o^2} \qquad (12)$$

where $N$ is the total number of windows in the domain, depending on the size of this window. This score lies between 0 and 1 and is typically computed for a large number of window sizes and plotted as a function of these window sizes. It allows to 270 determine the scale at which the target skill is reached. According to Skok and Roberts (2016)'s conclusions, the target skill can be set at 0.5 if the frequency of events is smaller than 20 % over the domain, which is our case. This target scale is the scale at which the forecast can be considered skillful and therefore 'useful' (Mittermaier and Roberts, 2010). To obtain the mean FSS over consecutive hours, the mean values of the numerator and denominator of the fraction are calculated individually and injected in equation 12.

## 3 Results for 10 min FEA

The performances of the proxies to predict lightning individually or in combination have been evaluated for the three validation days (7, 8 and 9 August 2018, Table 1). On the first day, 7 August 2018, convection started at 12 UTC with several convective cells located in the Pyrenees and from Brittany to Normandy (see Figure 2 for the locations). It intensified during the day, with hail and wind gusts associated with this convection. Ground level wind speeds of $112\ \mathrm{km\,h^{-1}}$ were measured at Cape 280 Bear, east of the Pyrenees. In the end of the day, from 17 UTC, thunderstorms are also observed in central France, moving


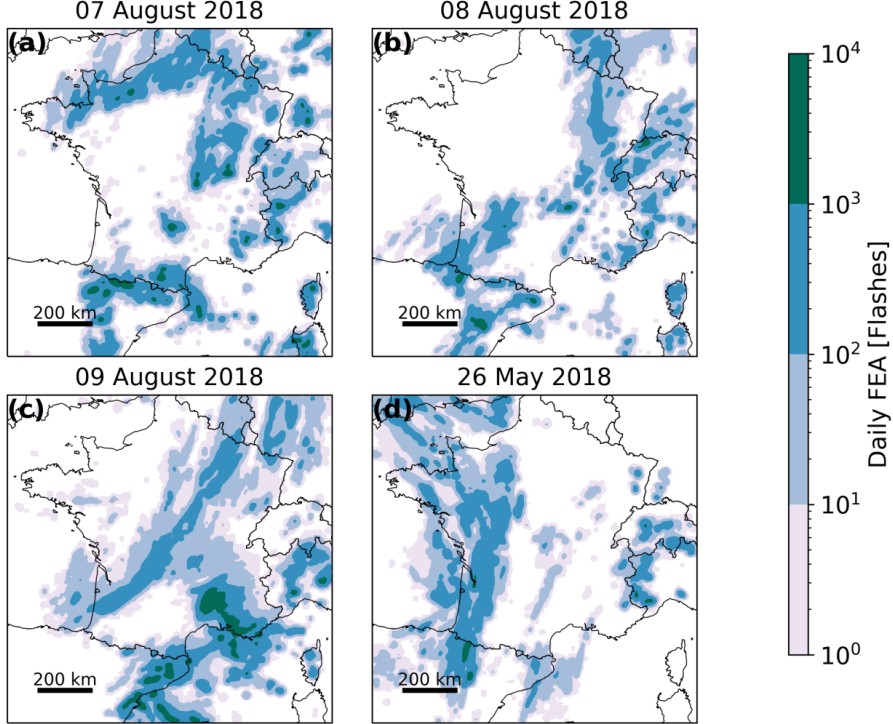

**Figure 5.** Daily FEA for (a), (b) and (c) 7, 8 and 9 and August 2018 and (d) 26 May 2018.

north-eastward, and in the Alps as well. FEA accumulated over the day is shown in Figure 5. On the second day, 8 August 2018, several small storms scattered over the Alps and Corsica appeared at 13 UTC. The lightning activity generally extended in the south-east region. From 18 UTC, lightning activity started west of the Pyrenees and propagated north-eastward during the night of 8 to 9 August. A convective system formed above the Cévennes–Vivarais region around 03 UTC on 9 August

and remained active all morning. It moved towards the Mediterranean coast in the afternoon. This storm was associated with intense precipitation (101.3 mm were measured at Aubagne at the end of the day) and a high density of lightning strokes, reaching values higher than 6000 flashes accumulated over the day (see Figure 5c), on a 7 km × 7 km grid.

### 3.1 Univariate models

The results of the fitting on the training set for all the regression models are presented in Figure 6 and Figure 7 for the graupel

mass and wmax, respectively (other proxies not shown here). Because data from the all the training days are concatenated and sorted, scatter plots of FEA versus each proxy are monotonic increasing curves. The lower FEA values, below 100 flashes, are well fitted with all the regression models for the graupel mass and wmax, but their predominance in the dataset (see colour shades in Figures 6 and 7) weighs the regression curves of the LinReg and LinSVR models too heavily. In consequence, the higher FEA values are not well fitted by the LinReg and LinSVR models, resulting in lower $R^2$ scores. All the scores for


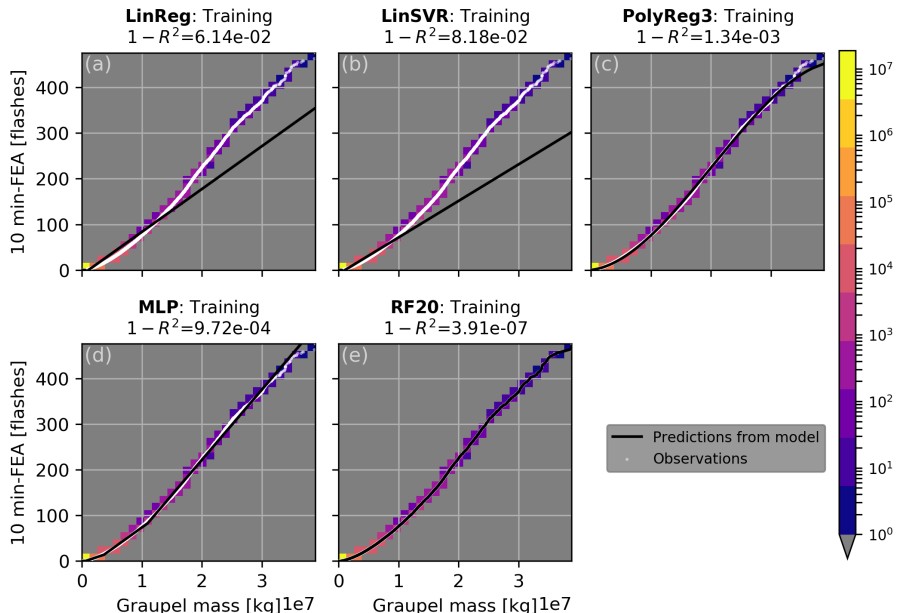

**Figure 6.** Model training curves in black for (a) linear regression, (b) linear support vector machine, (c) cubic polynomial regression, (d) multi-layer perceptron and (e) random forest regression overlaid on the global sorted distribution of observed FEA versus simulated graupel mass in white. The colour shades represent the number of points in each bin of $0.13 \times 10^7$ kg width.

**Table 2.** $1 - R^2$ score for each regression model and each proxy at the training.

| Proxy | PolyReg3 | MLP | RF20 | LinReg | LinSVR |
|---|---|---|---|---|---|
| RPC | $2.44 \times 10^{-2}$ | $1.71 \times 10^{-3}$ | $4.72 \times 10^{-7}$ | $3.36 \times 10^{-1}$ | $4.90 \times 10^{-1}$ |
| F1 | $2.92 \times 10^{-2}$ | $8.17 \times 10^{-4}$ | $1.90 \times 10^{-3}$ | $2.04 \times 10^{-1}$ | $5.62 \times 10^{-1}$ |
| F2 | $3.27 \times 10^{-3}$ | $1.52 \times 10^{-3}$ | $3.43 \times 10^{-4}$ | $1.20 \times 10^{-1}$ | $1.72 \times 10^{-1}$ |
| Graupel mass | $1.34 \times 10^{-3}$ | $9.72 \times 10^{-4}$ | $3.91 \times 10^{-7}$ | $6.14 \times 10^{-2}$ | $8.18 \times 10^{-2}$ |
| IWP | $5.97 \times 10^{-4}$ | $1.12 \times 10^{-3}$ | $3.56 \times 10^{-7}$ | $9.18 \times 10^{-2}$ | $1.31 \times 10^{-1}$ |
| LPI | $2.19 \times 10^{-2}$ | $1.08 \times 10^{-3}$ | $7.50 \times 10^{-4}$ | $1.80 \times 10^{-1}$ | $4.49 \times 10^{-1}$ |
| Updraft volume | $2.41 \times 10^{-2}$ | $4.17 \times 10^{-3}$ | $4.37 \times 10^{-7}$ | $2.21 \times 10^{-1}$ | $3.51 \times 10^{-1}$ |
| wmax | $1.39 \times 10^{-2}$ | $6.06 \times 10^{-3}$ | $1.93 \times 10^{-3}$ | $8.80 \times 10^{-2}$ | $1.31 \times 10^{-1}$ |

the training of each proxy with every regression model are summarized in Table 2. For all the proxies, the linear regressions LinReg and LinSVR present lower performances than the other three models, with $R^2$ scores ranging from 0.438 (LinSVR, F1) to 0.938 (LinReg, graupel mass). The MLP and RF20 models present $R^2$ scores very close to 1, systematically higher than 0.99 for all the proxies. Scores for the PolyReg3 range between 0.971 (F1) and 0.999 (IWP).



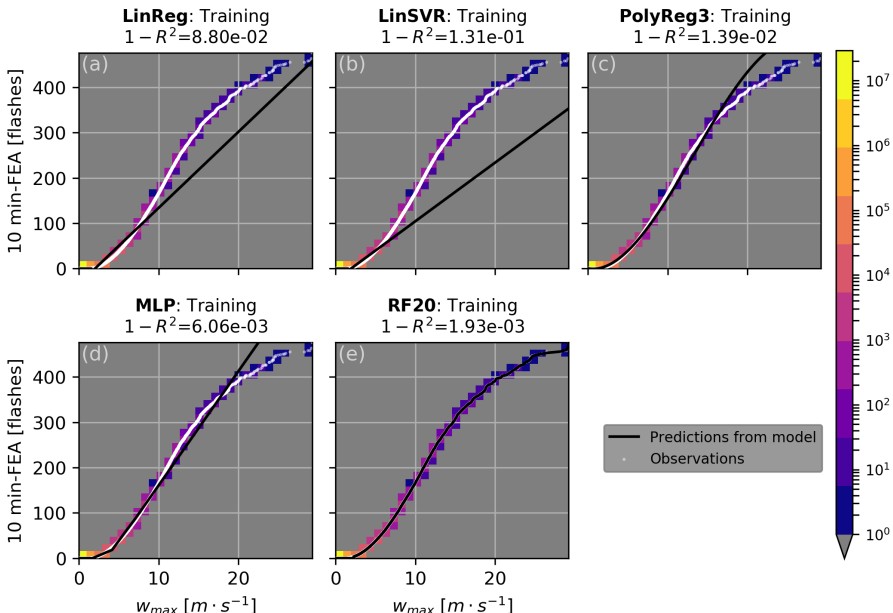

**Figure 7.** Same as Figure 6 but for the maximum vertical velocity (wmax) and number of points in bin of $0.974\,\mathrm{m\,s^{-1}}$.

The RF20 regression algorithm being the one presenting the best training performances as shown in Table 2, only the

validation for this model is shown in Figure 8 for the sorted distribution over the 3 days of validation for the graupel mass (a) and wmax (b). Other proxies are not shown here because they present very similar curves. Inspection of Figure 8 suggests a good fit for graupel mass and wmax since the predicted FEA are very close to the observations and this is confirmed by a $R^2$ score higher than 0.94 for wmax and higher than 0.98 for the graupel mass. Overall, the lowest score at the validation is 0.467 and is obtained for the updraft volume and the highest is for the IWP with $R^2 = 0.989$.

A good fit on the validation set is not enough to conclude on the ability of a proxy to predict lightning at the right position and time. In consequence, the FEA is computed for each hour for a given proxy of the validation dataset and compared to the observed FEA using FSS. As stated in section 2.2, the proxies are calculated using 1-hour AROME-France forecasts from the assimilation cycle. Figure 9 shows a typical example of the observed and computed 10 minutes FEA fields using the RF20 regression model with the graupel mass, a microphysical proxy, for 9 August 2018 at 13 UTC. The minimum observed FEA

value is 1 flash and the colormap is in log scale. The lightning activity observed in the North-East of Spain is well reproduced by the regression model, as well as the storms above South-East of France, although slightly shifted northwards because of the displacement error of the graupel mass forecast from the NWP system. On the other hand, the lightning flashes observed above Corsica are not well reproduced by the RF20 regression model even if graupel has been simulated. The FSS is calculated for two different thresholds: 1-flash threshold includes all the observations and predictions with values equals or higher than 1

flash and 10-flash threshold is stricter and indicates the ability of the regression model to reproduce the high values of FEA. The lower threshold is often associated to a greater FSS since it implies larger areas that are less prone to displacement errors.


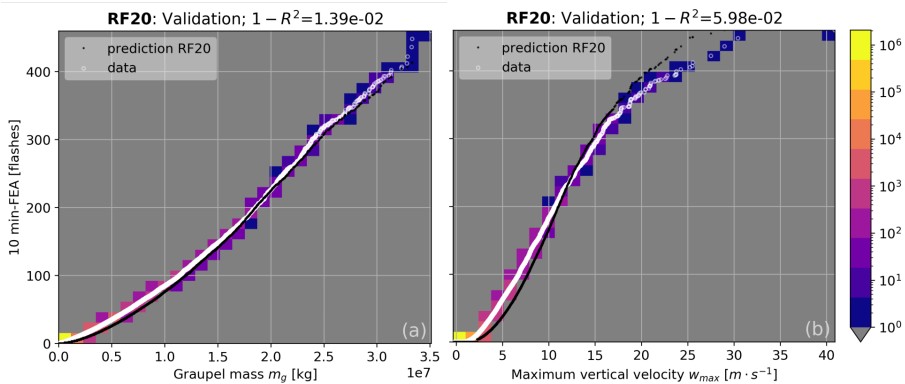

**Figure 8.** Sorted distribution for the validation set of the observed FEA versus the simulated graupel mass (a) and maximum vertical velocity (b) in white. Black dots correspond to the FEA predicted by the RF20 regression.

For this example, the 1-flash FSS is particularly high, already higher than the FSS target at the smallest neighborhood size meaning that the forecasts, both the NWP forecast and the prediction from the regression model, are already useful at the FEA scale.

Similarly to Figure 9, Figure 10 presents the results for the FEA modelled with the RF regression model using wmax, which is a dynamical proxy. In Figure 10b, only the maximum vertical velocities higher than $1\,\mathrm{m\,s^{-1}}$ are shown. The FSS indicates a less successful prediction for the FEA than with the graupel mass and this conclusion is supported by a visual inspection of the structure of the modelled FEA in Figure 10c. The FEA built from wmax presents isolated structures with a lower spatial coverage than the FEA computed from the graupel mass, resulting in less coincident spatial overlaps between observed and simulated FEA.

This difference of structure in the modelled FEA can be observed for all the microphysical proxies versus the dynamical ones, and for any validation hour. Indeed, the FEA modelled with either the IWP or F2 presents very similar results compared with the graupel mass both in terms of spatial coverage and FEA amplitude, for a given time (not shown). In contrast, FEA modelled with F1 and the updraft volume is scattered on the map in a similar pattern to FEA computed with wmax, as in Figure 10c. This areal coverage difference between flash extent predicted with either F1 or F2 was already emphasized by McCaul et al. (2009). FEA modelled with the rimed particle column, identified here as a microphysical proxy, presents also the same characteristics as with the graupel, i.e. in terms of spatial coverage and amplitude. Interestingly, when the observed FEA is plotted as a function of the rimed particle column thickness for the training distribution, the curve presents a net increase in FEA when the RPC reaches $3500\,\mathrm{m}$, a close value to the $3000\,\mathrm{m}$ described in Figueras i Ventura et al. (2019), showing consistency between both our studies. The LPI is more difficult to classify since it is a product of both microphysical and dynamical terms. However, the FEA modelled using the LPI is very similar to the one obtained with wmax. In consequence, we consider the LPI as a dynamical proxy.

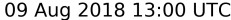

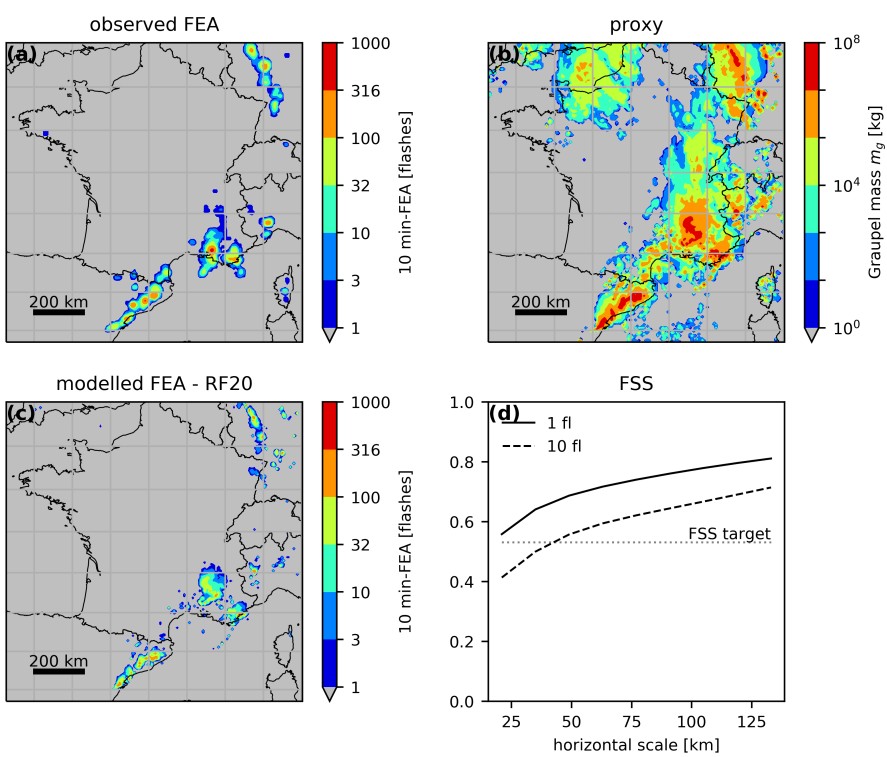

**Figure 9.** Spatial distribution over the studied domain for 9 August 2018 of (a) the observed FEA between 12:55 and 13:05, (b) AROME-France simulated columnar graupel mass valid at 13 UTC, and (c) RF20-based modelled FEA for 12:55 to 13:05 using the graupel mass. Panel (d) shows the FSS for the observed versus modelled FEA plotted as a function of the neighborhood size.

The examination of the FSS calculated for the whole validation period for FEA predicted with the RF20 model (Figure 11) highlights the difference of performances between these two types of proxies. The microphysical proxies systematically score

better than the dynamical ones for the two different thresholds. It is however difficult to conclude on the best proxy among the IWP, the graupel mass, F2 and the rimed particle column because the differences are not significant as the confidence intervals overlap (not shown here). The FEA forecasts beyond 1 flash obtained with the microphysical proxies require neighbourhood sizes ranging between 50 and 70 km to reach skillful spatial scales, whereas the target skill is never reached whatever the range for the dynamical proxies-based FEA.

Although results were depicted mainly for the RF20 model in this section, the FEA modelled with the other regression models presents very similar characteristics, with FEA amplitudes slightly lower with LinReg and LinSVR. For example, for the same situation as above, 9 August 2018 at 13 UTC, domain-wide maximum modelled FEA values are 174, 204, 265, 269 and 269 flashes for LinSVR, LinReg, MLP, PolyReg and RF20 regression models, respectively, for an observed value of 412

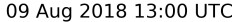

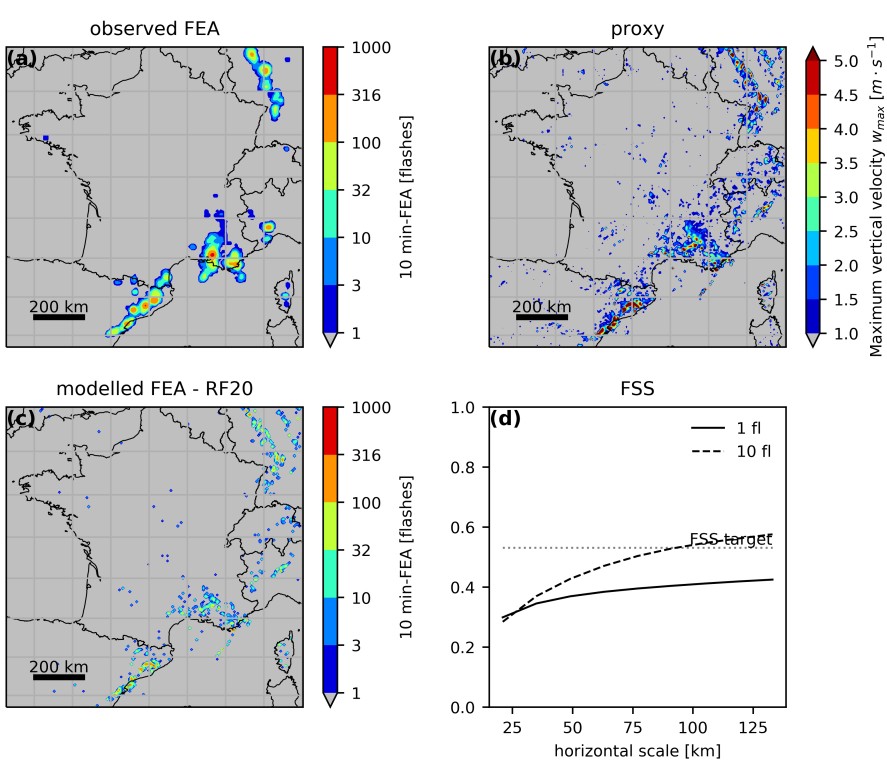

**Figure 10.** As in Figure 9 but with the maximum vertical velocity, wmax.

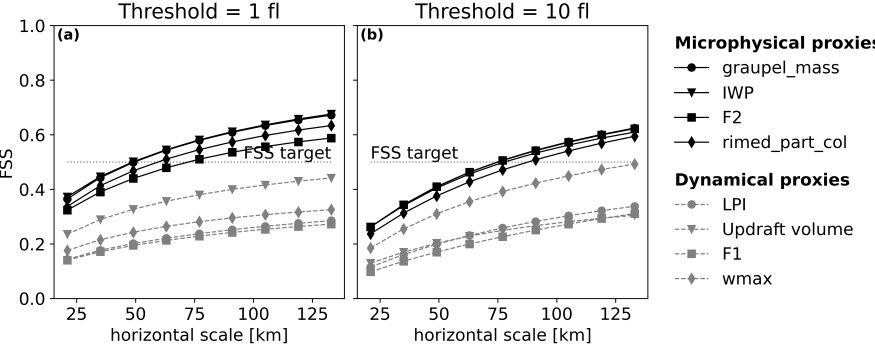

**Figure 11.** Mean FSS of the whole validation period for all the proxies for (a) 1-flash threshold and (b) 10-flash threshold, for RF20-based FEA.

flashes. The general underestimation of domain-wide maximum FEA occured for this specific case, however, we do not see a

general negative bias of FEA maximum.


## 3.2 Multivariate models

The objective here is to take advantage of the information contained in several proxies by combining them into a multivariate regression model. A problem arose here because of the high correlation between some of the proxies, called features in ML, as derived from the same fields from the AROME forecast. The multicollinearity among the features is demonstrated when

trying to assess the importance of each feature using the Permutation Feature Importance method (Breiman, 2001). The goal of this method is to determine how much the performance of a model relies on a single feature by calculating the difference between the $R^2$ score from the original dataset for a given regression model and the score when one feature of the dataset is randomly shuffled. If the shuffled feature is important, the score difference should be large. However, when the Permutation Feature Importance is calculated on our training dataset, the score differences for the random forest regression model are all

smaller than 0.1 (not shown) which suggests that none of them are important. This is in contradiction with the results presented in section 3.1 with high skill for most proxies, meaning that when a feature is corrupted, very similar information can be found in another one. Multicollinearity is a problem because it decreases model accuracy and robustness and increases overfitting (Cohen et al., 1983; Neter et al., 1996; Chatterjee and Hadi, 2006). To avoid these issues, we went through a process of features selection using two different methods: i) hierarchical clustering on the Spearman correlation-rank ordered features to study

their correlation and drop redundant variables, and ii) a principal component analysis (PCA).

First, the correlations between the proxies are examined more profoundly by measuring their Spearman correlation coefficients. This coefficient is a non-parametric measure of the rank correlation between two variables. The more the variables are monotonically related the higher the coefficient is. The coefficient values for the studied proxies are displayed on a dendrogram and a correlation matrix in Figure 12. Overall, the dendrogram reveals two main clusters: one composed of the IWP, F2,

the graupel mass and the LPI and another one that contains the updraft volume, wmax, F1 and the RPC. Color shades of the correlation matrix highlight the fact that the members of the first cluster, IWP, F2, graupel mass and LPI, present a stronger correlation than any member of the second cluster. As expected, the IWP and F2 are the most correlated features, because they are deduced from practically the same AROME fields: the difference lies in the mixing ratio of ice taken into account in F2 and an integration column smaller for the IWP. Surprisingly, the RPC is more correlated with F1, wmax and the updraft volume

than with the other microphysical proxies and the LPI is closer to the graupel mass than to wmax. The objective is then to select one feature per cluster and fit a random forest regression algorithm with them. This time, a forest of 64 trees was grown and the depth of the trees was set to 80. According to Oshiro et al. (2012), a forest of 64 to 128 decision trees is recommended to reach a good balance between processing time, memory usage and performances. The depth of tree was set to 80 because no improvement in performances was observed beyond, only an increase in processing time. Several proxy combinations were

tested and the best results were obtained when graupel mass and wmax were selected. Figure 13a shows the FEA modelled when the RF algorithm is trained with these two proxies, for the same validation hour as Figures 9 and 10, i.e. on 9 August 2018 at 13 UTC. The general areal coverage is very similar to what was predicted with the graupel mass alone, especially in South of France and North-East of Spain, with some additional isolated patches that can be attributed to the contribution of wmax. However, they do not improve significantly the overall prediction compared to the graupel mass alone and this is also
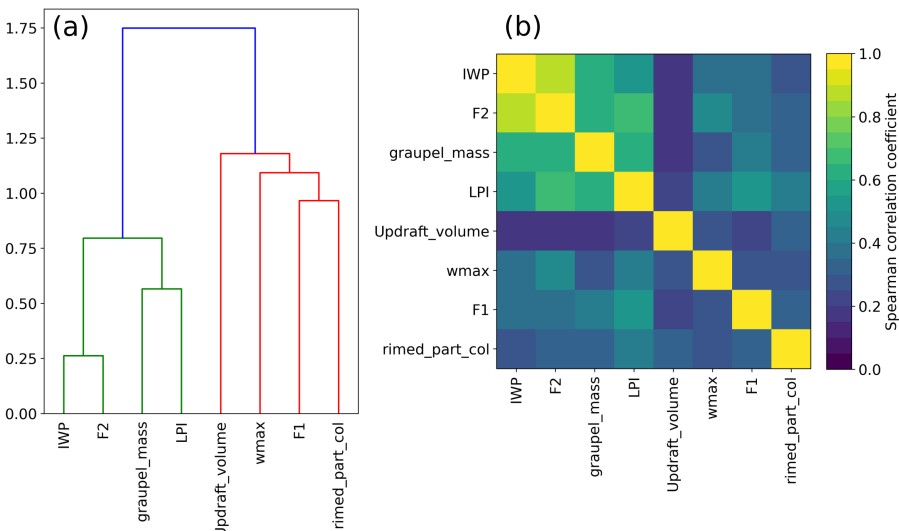

**Figure 12.** (a) Dendrogram of the Spearman-rank order correlations of the proxies and (b) correlation matrix.

demonstrated with the FSS for the whole validation period shown in red in Figure 14. Other combinations were tested, for
instance using the updraft volume and the IWP, for which the FSS is plotted as well in Figure 14, but the performances were
all quite similar.

The PCA is a dimension reduction technique that creates new uncorrelated variables with linear combinations of each feature
from the original dataset (Jolliffe and Cadima, 2016). The linear coefficients are called the principal component (PC) loadings
and are calculated such as the PCs have successively a maximum variance. The proportion of total variance explained by each
PC is often expressed as a percentage and only the first PCs that cumulatively explain 70 % of the total variance are kept
(common cut-off point according to Jolliffe and Cadima, 2016). In our case, the PCA is applied to the non-sorted standardized
(mean equals to 0 and standard deviation to 1) array of stacked column-wise features. Analysis of the cumulative explained
variance ratios (not shown) reveals that only the first two PCs are required to describe at least 70 % of the variance. The random
forest regression algorithm is then fitted with the selected PCs, that were sorted beforehand. The same transformation to a
dimensionally reduced dataset is applied to the validation dataset. The FEA is then calculated for each hour of the validation.
Figure 13b depicts the FEA modelled using the first two PCs for 9 August 2018 13 UTC. It presents little to no difference with
the FEA modelled using simultaneously graupel mass and wmax, highlighting the contribution from both microphysical and
dynamical proxies. The FSS for multivariate models using either the first two PCs or the first five PCs are plotted in Figure 14
as well. Overall, they do not present a clear improvement compared to microphysical proxies alone.

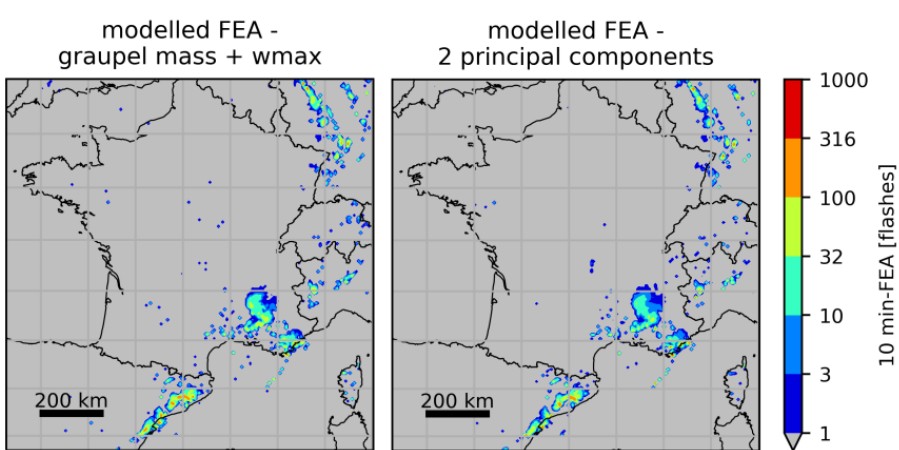

**Figure 13.** FEA modelled using (a) graupel mass and wmax simultaneously and (b) the first two principal components of a dataset composed of all the proxies, for 9 August 2018 at 13 UTC.

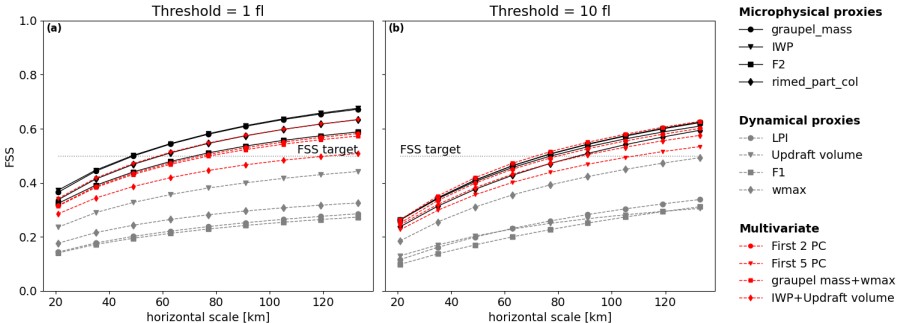

**Figure 14.** Same as Figure 11 but the FSS for individual proxies are overlaid with the FSS for some multivariate models in red.

## 4   Comparison with the McCaul et al. (2009) calibration

The objective of this section is to compare the above introduced calibration method, where regression functions are calibrated with concatenated and sorted data from the training dataset, to the one developed by McCaul et al. (2009). Indeed, their lightning forecasting algorithm is incorporated in several resolved-convection forecast NWP systems, for example WRF (McCaul et al., 2020) and often used as a comparison for lightning forecasts (e.g. Lynn, 2017). To follow a similar method, only the domain-wide maxima of the FEA observations and of simulated F1 and F2 are extracted for each day of the training set and plotted in Figure 15. The FEA observations present a moderate correlation with the proxies, with Pearson correlation coefficients $r = 0.70$ for F1 and $r = 0.74$ for F2. The linear best-fit curves obtained with the reduced major-axis regression (Figure 15) have slopes of $7231 \, \mathrm{fl\,m\,s^{-1}}$ and $10.14 \, \mathrm{fl\,kg^{-1}\,m^2}$ for F1 and F2 respectively, with an intercept forced at the origin. Hence,


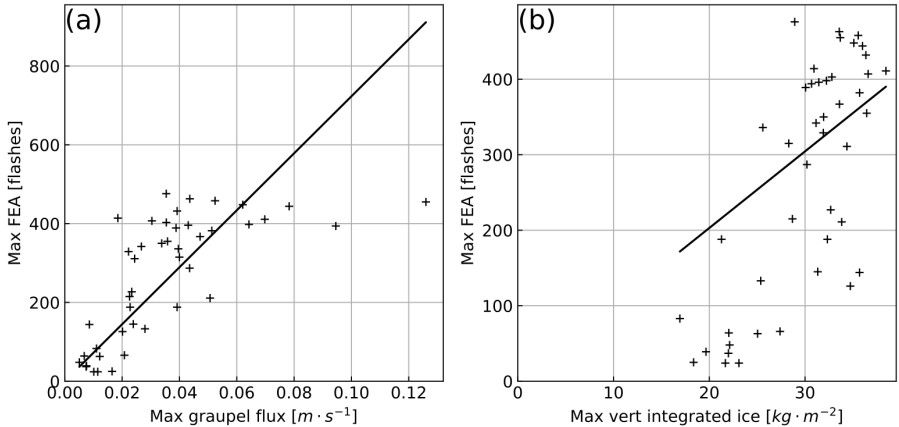

**Figure 15.** Domain-wide maximum observed FEA versus (a) simulated F1 and (b) simulated F2 with regression curves obtained with the reduced major-axis regression technique.

the blended function F3 described by equation 9 can be written as

$$F3 = 0.95(7231 \cdot F1) + 0.05(10.14 \cdot F2) \tag{13}$$

The 10 min FEA is modelled for every hour of the validation period successively with F1, F2 and F3. An example for the 9 August 2018 at 13 UTC is shown in Figure 16. From a visual inspection of Figure 16b and Figure 16c, the same conclusion as McCaul et al. (2009) can be drawn regarding the areal coverage of the FEA modelled with F1 and F2 individually: while F2

widely overestimates the areal coverage of the FEA, F1 presents scattered and isolated features and an areal coverage lower than the observed one. The blended approach seems to reproduce the observed areal coverage better, in a similar pattern to what was obtained with the microphysical proxies with our method. However, the amplitude of the FEA is underestimated, with maxima reaching less than 32 flashes whereas the observation reaches values beyond 400 flashes for this specific example. This underestimation of high values is highlighted by the FSS for the whole validation period, shown in Figure 17. For F3-FEA, the

10-flash FSS is very low, lower than 0.2 for horizontal scales up to 130 km, implying that only few values beyond 10 flashes are modelled compared to what is observed. It means that taking only the maximum values to fit the regression function, as in McCaul et al. (2009), is not enough to represent the dataset variability.

Ultimately, the F3 function presents a simple empirical way to forecast lightning density using both microphysical and dynamical properties of the cloud and was able to reproduce the areal coverage quite well overall for the validation period.

However, the predicted F3-FEA values have too low amplitude whereas our method, where all data from the training dataset are used to fit the regression models, produces FEA with values closer to the observed ones.

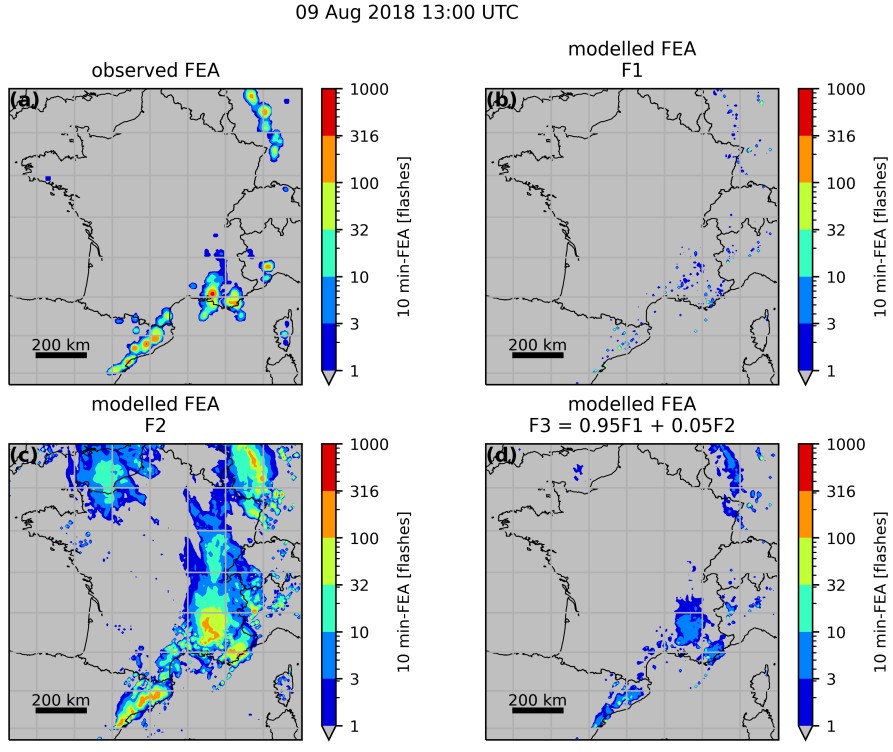

**Figure 16.** Observed FEA for 9 August 2018 at 13 UTC (a) and FEA modelled with F1 (b), F2 (c) and F3 (d).

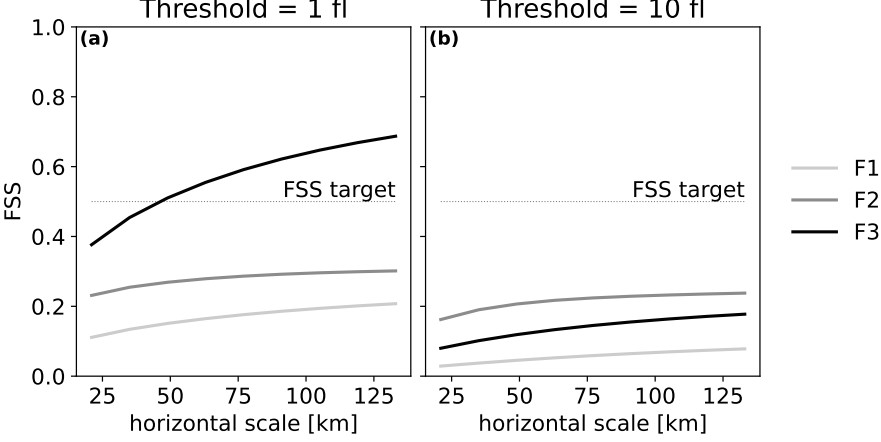

**Figure 17.** FSS for the whole validation period for FEA modelled with F1, F2 and F3 for two different thresholds.




## 5 Sensitivity to the FEA accumulation period

The sensitivity of the relationship between a proxy and observed FEA to the accumulation period of the FEA has been evaluated. In the following, the graupel mass is the proxy selected to perform the study because it is one of the proxies presenting the strongest correlation with the FEA ($R^2 > 0.9$ for all regression models on the validation set). Only the cubic polynomial regression model is considered for this sensitivity study.

The method to calibrate the regression functions is quite similar to the one described in section 2.4: graupel mass is extracted from the 1 h AROME forecasts from the assimilation cycle and FEA accumulated successively on 5, 10, 15, 20, 30 and 60 minutes are generated for every day of the training dataset (44 days). The graupel mass is not accumulated, it is an instantaneous data field, valid at time $t$. The FEA accumulation period is always centered on the corresponding time $t$ of the AROME forecast, with a time step of 5 minutes. For instance, the 10 min FEA is accumulated between $t - 5$ min and $t + 5$ min but the 15 min FEA is accumulated between $t - 5$ min and $t + 10$ min. Data from the whole training dataset are concatenated, flattened and sorted. Points where both graupel mass and FEA are null are removed and model fitting is performed on normalized data. To compare the observed and modelled FEA for different time accumulations, they are all divided by their respective accumulation period to obtain comparable products expressed in $\mathrm{fl\,min^{-1}}$ and hereafter called Flash Extent Rate (FER).

The date of the 26 May 2018 was selected to study the influence of the different accumulation period. On this day, a series of storms formed north of Spain, west of the Pyrenees at 08 UTC. These cells propagated northward, following the Atlantic coast, arriving above Bordeaux at 11 UTC. At around 13 UTC, these systems evolved into a bow echo. The bow echo continued its route toward Normandy and Great Britain and left our domain of interest at 22 UTC. For a more detailed description of the event, see Mandement and Caumont (2020).

This event was chosen because of its high propagation speed, approximately $50\,\mathrm{km\,h^{-1}}$ on average, which should emphasise the effects of the various accumulation times on the performances of the graupel mass-based FEA model. Figure 18 is an example presenting observed FER for four different accumulation times, 5, 10, 30 and 60 min, all centered on 26 May 17 UTC. One can notice the spreading of FER areal coverage northward and southward when the accumulation time increases. Only the western half of the domain is considered (grey background in Figure 18) in order to focus on the bow echo event and avoid other lightning activity to mislead the calculation of the FSS.

The polynomial regression curves obtained from the training set with various accumulation times are plotted in Figure 19a. For graupel mass values lower than $0.5 \times 10^7$ kg, the regression curves are coincident, meaning that there would be no difference in the predicted FER with various accumulation times for those graupel mass values. The study of the distribution of the non-zero graupel mass values for the time period of interest, 13 UTC to 22 UTC on 26 May 2018 (Figure 19b), indicates a large predominance of low graupel mass values, with 99.7 % of them being lower than $0.55 \times 10^7$ kg. In consequence, the graupel mass values that would imply a variation in the modelled FER for the different accumulation times are hardly reached, resulting in very similar FER. Those resemblances are highlighted by the FSS, plotted for the bow echo lifetime period, 13 UTC to 22 UTC, in Figure 20. Thresholds are set at $0.1\,\mathrm{fl\,min^{-1}}$ and $1\,\mathrm{fl\,min^{-1}}$, to be consistent with previous sections FSS thresholds set at 1 fl and 10 fl for an accumulation time of 10 min. For both thresholds, the FSS of the FER with various

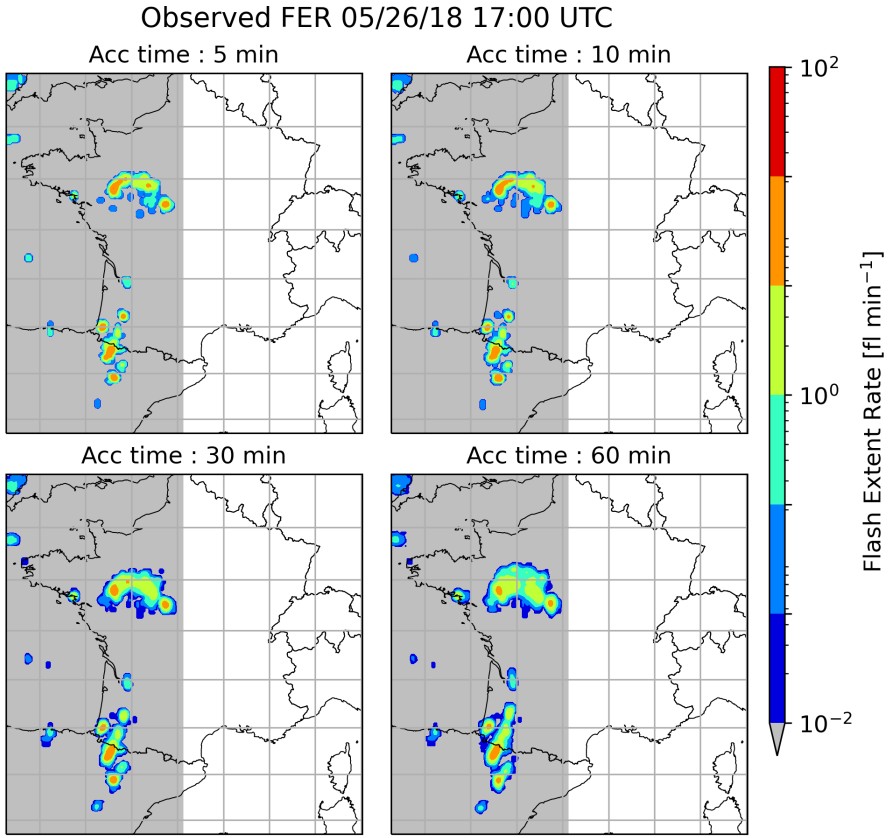

**Figure 18.** FER observed on 26 May 2018 at 17 UTC for various accumulation times.

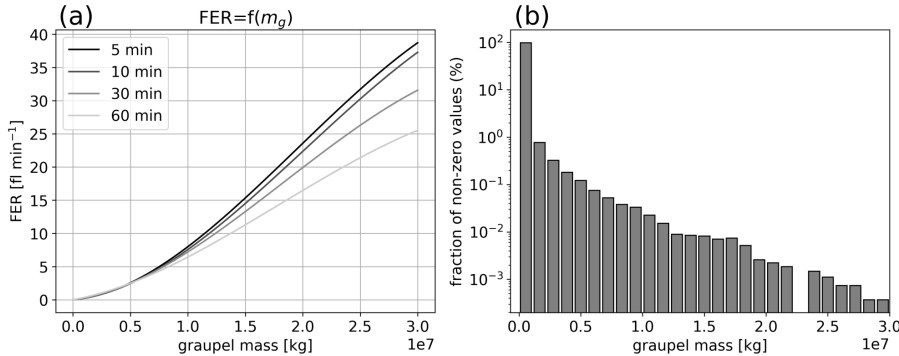

**Figure 19.** (a) Regression curves for various accumulation times and (b) distribution of the non-zero graupel mass values simulated between 13 UTC and 22 UTC on 26 May 2018.


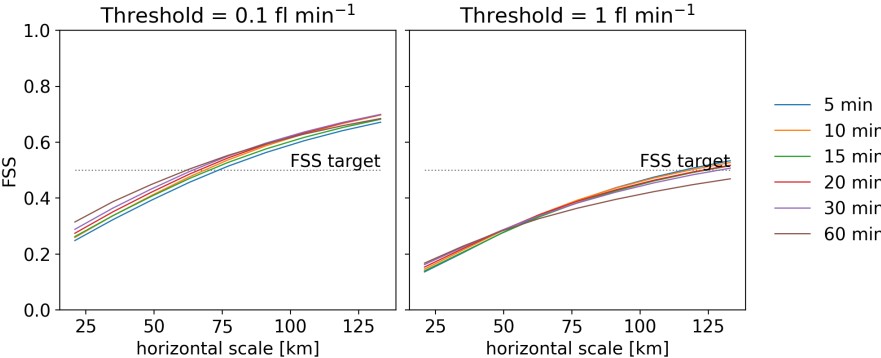

**Figure 20.** FSS for the bow echo lifetime period, 13 UTC to 22 UTC on 26 May 2018, for all the accumulation times tested, with two different thresholds.

accumulation times are very close. The 90 % bias-corrected and accelerated (BCa) bootstrap confidence intervals (see Efron and Tibshirani, 1993) for each threshold and each accumulation time overlap (not shown), meaning that the differences between the FSS are not significant.

This sensitivity study did not allow the identification of an optimal accumulation period. This leads to think that the choice

of the accumulation period of the flash extent is not of great importance, as long as it centered on the prevision time. Therefore, the accumulation period can be chosen by the user depending on the available observations or the intended usage.

## 6   Discussion and conclusion

In preparation for the upcoming launch of the new generation European geostationary satellites with LI onboard, a calibration between pseudo MTG-LI observations and simulated cloud parameters was introduced in this study. This calibration

was performed in order to find the best proxy variables for an observation operator in the prospect of satellite LDA in the new AROME-France 3D-EnVar assimilation scheme, but possible applications could also include AROME-related lightning diagnostics.

First, a set of proxy variables was selected based on their performances to predict lightning as demonstrated in several studies. Empirical relationships between simulated proxies and FEA observations were established climatologically by adjusting

different regression models to concatenated and sorted data from 44 days of year 2018. It means that the climatology of the FEA predicted with those regression functions are set to the climatology of the observed FEA, allowing unbiased NWP system equivalent, which is useful in a context of data assimilation. The regression functions were used on 3 validation days to compare the predicted FEA using the different proxies. It was shown that the microphysical proxies were the most successful to reproduce FEA areal coverage and amplitude. It is however difficult to tell apart the different microphysical proxies because

they all present very similar performances to forecast lightning, with FSS very close to each other. In the context on LDA, we recommend the use of a microphysical proxy to build the observation operator, with a preference for proxies relying on several





variables, IWP or F2 for instance, because when assimilated, a change in the proxy will have an impact on several prognostic variables of AROME-France.

In the context of 3D-EnVar LDA, a linearized version of the observation operator is required. While the RF20 regression
model is the one presenting the best performances, it is not the simplest relationship to differentiate because it resembles a flowchart rather than a continuous function. The cubic polynomial regression, as the best compromise, exhibits performances only slightly below this more complex model and is still easily differentiable. In consequence, polynomial regression is better suited for LDA and will likely be used in future work. To build a LDA observation operator for another NWP system, the calibration of the regression coefficients will have to be performed again with FEA observations and proxies simulated with
this NWP system but the same method can be applied to process the data climatologically, i.e. selecting a sufficient amount of days to sample the data, and concatenating and sorting them.

Secondly, it was shown that combining proxy variables into multivariate models does not improve FEA prediction overall, implying that adding dynamical variables is unnecessary to forecast FEA. Several proxy combinations were tested, selected to avoid redundancy in the dataset, but none of them presented better FSS than microphysical proxies alone.

Thirdly, results from an existing lightning calibration from the literature (McCaul et al., 2009) were compared to the hereby calibration. Using a similar methodology to adjust regression functions and to deduce the calibration parameters, the predicted FEA for the validation days exhibits amplitudes largely inferior to the observed one but with an areal coverage similar to what is obtained with the microphysical proxies alone.

Finally, sensitivity experiments on the FEA accumulation time were performed. The few differences among the regression
functions implied that the modelled FER would likely not differ much according to the accumulation time, meaning that there is some flexibility in the accumulation time choice, depending on the user's preferences or operational constraints.

*Author contributions.* This work was carried out by PC as part of her PhD thesis under the supervision of OC, ED and MM. FE's python codes were made available to PC and she used and adapted certain parts of these codes. PC wrote the paper and created all the plots. All authors contributed to the revisions of the paper.

*Competing interests.* The authors declare that they have no conflict of interest.

*Acknowledgements.* We thank Météo-France and the Occitanie Région for funding the PhD. This work is part of the SOLID project (Space-based Optical LIghtning Detection) supported by CNES. We also wish to thank Isabelle Couasnon and Tom Nicolau for providing the Météorage data.





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
