# Peer review of "An observation operator for geostationary lightning imager data assimilation in the French storm-scale numerical weather prediction system AROME"

_Natural Hazards and Earth System Sciences, 2022_

## Author Comment (AC1)

**Replies to comments of Reviewer #1:**

I have read this paper and find it very comprehensive, clear, well written, and important in the field of predicting lightning hazards using NWP models, and the future use of geostationary satellite lightning data that will soon become available over Europe with the launch of the MTG LI sensor. I have a few minor comments related to clarifications, language, and methodology.

We thank Prof. Colin Price for his review and comments. Our responses are as follows (in black), in order of the written comments (in red). All changes and additions will appear in the revised version. The line numbers mentioned in the answers refer to the revised version of the manuscript.

1) Abstract and title. From the abstract and title, it is not clear if the paper is about a) improving lightning assimilation into NWP models, in order to improve forecasts, or b) to understand what NWP parameters are best related to lightning. While these are connected, it is somewhat confusing in the abstract. Maybe this could be clarified in the abstract, and maybe the title too.

We reformulated our title to «A satellite lightning observation operator for storm-scale numerical weather prediction». The abstract has been shortened and the objectives clarified. Below is the revised version of the abstract:

*This study aims at simulating satellite-measured lightning observations with numerical weather prediction (NWP) system variables. A total of 8 parameters, calculated with the AROME-France NWP system variables, were selected from a literature review to be used as proxies for satellite lightning observations. Two different proxy types emerged from this literature review: microphysical and dynamical proxies. Here, we investigate which ones are best related to satellite lightning and calibrate an empirical relationship between the best parameters and lightning data. To obtain those relationships, we fit machine learning regression models to our data. In this study, pseudo flash extent accumulation (FEA) observations are used because no actual geostationary lightning observations are available yet over France and non-geostationary satellite lightning data represent a too small sample for our study. The performances of each proxy and machine learning regression model are evaluated by computing Fractions Skill Scores (FSS) with respect to observed FEA and proxy-based FEA. The present study suggests that microphysical proxies are more suited than the dynamical ones to model satellite lightning observations with the AROME-France NWP system. The performances of multivariate regression models are also evaluated by combining several proxies after a feature selection based on a principal component analysis and a proxy correlation study but no proxy combination yielded better results than microphysical proxies alone. Finally, different accumulation periods of the FEA had little influence, i.e., similar FSS, on the regression model's ability to reproduce the observed FEA. In future studies, the microphysical-based relationship will be used as an observation operator to perform satellite lightning data assimilation in storm-scale NWP systems and applied to NWP forecasts to simulate satellite lightning data.*

2) line 47: FEA: Why is this called extent if you are talking about the flashes per pixel, and the pixels are of constant size?

The pixel size of geostationary lightning imagers is a few kilometers wide (7 km x 7 km for MTG-LI at Western Europe latitudes) which can be smaller than the size of a flash. Thus, the number of illuminated pixels will depend on the actual size/extent of the flash and scattering of the light by the cloud microphysics. One flash often illuminates more than one pixel, and counting (=*accumulating*) all the flashes, one by one, with channels propagating and light scattering through a certain number of pixels, yields the FEA.

3) line 67: Why were only 1 hour forecasts tested? Why not 3 hours, 6 hours?

In data assimilation, the analysis, i.e. the best estimate of the state of the atmosphere, is obtained using information from the background (the latest forecast of the model) and the observations. This analysis will be the starting point of the new forecast. In AROME-France, the period of the data assimilation cycle is 1 hour and the background is the forecast from the previous hour. During the data assimilation process, the observation operator will be applied to this background to obtain model equivalent. So we calibrate the FEA-proxy relationship with the 1 hour forecast outputs because this relationship will be specifically applied to these forecast outputs.

The 1 hour forecast is also the one carrying the least forecast errors because it avoids the spin-up phenomena (large forecast errors in the first tens of minutes due to an imbalance in the model fields) and limits the model errors that increase after a few forecast hours. This justification has been added line 66.

4.1) line 90-91: Are 10 days really enough for making such a conversion between CG to total lightning?

The NLDN detects both CG and IC discharges (this information has been added in the revised manuscript, line 83) so the generator was trained with total lightning data. No conversion between CG and total lightning was needed. The 10 days were selected based on the NLDN lightning activity within the study domain. We agree that a more comprehensive training dataset would in general, as always, stabilize the statistical characteristics towards a full representation of the population, not just a sample. However, due to the sheer amount of lightning captured by GLM over the region during one day, using more than 10 days for the training was computationally too expensive. The 10 days were selected in different seasons and for different lightning-producing phenomena, e.g., squall lines, MCSs, thermal free convection. Full 24-hour periods were used to include both day and nighttime GLM records on each day. This question is discussed in detail in Erdmann et al. (2022).

4.2) Are the storms in the SE USA similar to the storms in France?

We addressed this question when building the generator. ISS-LIS records where analyzed in a statistical sense over both the region in the SE USA and over France and used as a common reference to compare NLDN and Meteorage lightning data. It was found that the distributions of flash characteristics from these sets of lightning records show similar shapes and statistics. The analyses included the number of events per flash (IC/CG strokes per flash for NLDN and

Meteorage), flash extent, flash duration, and mean and maximum optical radiance (LF current for NLDN and Meteorage) of a flash. In addition, relative flash detection efficiency was calculated for all instruments. For details about this comparison, please see F. Erdmann's PhD manuscript (Erdmann, 2020, chapter II.2.4).

4.3) How do you think these issues may have influenced your results?

We are convinced that the data generator used to generate MTG-LI pseudo-observations for this study is the best and most realistic tool of its kind. At this point, it is difficult to estimate the true quality of the generated MTG-LI records. We have already planned a comparison to real MTG-LI observations, as soon as they are available, to assess the quality of the pseudo-observations.

5) line 96: Figure 1

It has been modified, thank you.

6) line 129: I should point out that PR92 looked at CONVECTIVE cloud top height, not all cloud top height. Hence in the NWP it is possible to isolate convective cloud tops from other clouds (cirrus).

Yes, indeed. In Price and Rind (1992) you used the cloud top pressure and optical depth from satellite observations to determine whether a cloud is convective or not. But in AROME-France, deep convection is not parameterized so one cannot directly differentiate convective and stratiform areas. We thought of using updraft characteristics combined with a minimum threshold on specific contents of cloud droplets and ice crystals as a criteria to determine the convective cloud top height but it would have complicated the observation operator in a context of data assimilation.

7) line 137: detailed

It has been modified, thank you.

8) line 151: why are the updrafts so low in your model? This does not match reality in thunderstorms.

In our simulations, the vertical velocities in convective regions are roughly between 3 and 15 m/s in general but they can reach values higher than 40 m/s: see $x$ axis in panel (b) of Figure 8. What we meant here is that those occur in regions that have a very limited horizontal extension which does not pixel-to-pixel correlate well with flash extent observed from space. The proportion of data points with vertical velocities higher than 5 m/s is too low compared to the proportion of data points with a FEA higher than 0. It would mean that, statistically, FEA can be observed when there is no updraft volume with vertical velocities higher than 5 m/s. That is why we chose to use the updraft volume with vertical velocities higher than 1 m/s instead, in order to increase the number of updraft volume data points so that its horizontal extension matches better the horizontal extension of the FEA.

 It has been reformulated in the manuscript at lines 149-154 as such:

*In our AROME-France simulations, vertical velocities in convective regions are roughly between 3 and 15 m/s, even though some values can exceed 40 m/s. However, those values occur in regions with very limited horizontal extension, smaller than the FEA horizontal extension. To have a matching number of non-zero values of FEA and updraft volume, the updraft volume is here defined as the sum of grid cell volumes with vertical velocity higher than 1 m/s for each column from -5 °C to the roof. The lower velocity threshold compared to the literature is thus an adaption to our AROME-France model specifications.*

9) line 404: convection-resolved

It has been modified, thank you.

10) line 465: prevision time? Maybe prediction time?

We chose to change it to "analysis time" to be consistent with the explanation given line 446.

**Replies to comments of Reviewer #2:**

The authors describe a method for building a lightning observation operator for eventual use in assimilation of MTG LI flash extent accumulation data in the AROME NWP model. They use synthetic MTG LI data built on prior work. They reach a clear recommendation that is ready for use in preparing an operational modeling system for future observations. The practical aspects of the authors' study are relevant to global operational weather centers. The techniques used are comparable to current state of the art efforts.

Furthermore, the obvious difference in the spatial coverage of the so-called microphysical and dynamical proxies is a valuable scientific result, and one I was not expecting to see emerge from a model-observation comparison where the model does not include electrification process. I would interpret this result as telling us that the integrated action of all updrafts in forming ice-phase precipitation is more important than any one updraft, consistent with the patchy appearance of the dynamical proxies. And of course, ice-phase precipitation is widely accepted as the primary ingredient in thunderstorm electrification, so it makes physical sense from that point of view, too.

I commend the authors on a comprehensive study that is also very concisely and clearly written – especially for the large number of parameters compared and the number of methodologies employed. Below, I offer some minor comments that could help clarify a few details.

We thank Prof. Eric Bruning for his review and comments. His interpretation of our results has been added to the manuscript's conclusions at line 499. Our responses are as follows (in black), in order of the written comments (in red). All changes and additions will appear in the revised version. The line numbers mentioned in the answers refer to the revised version of the manuscript.

Line 31: The lightning jump is coincident or slightly lags some aspects of intensification (updraft volume, number concentration of precipitation in the mixed phased), but leads other intense phenomena at Earth's surface. Which kind of intensification is meant?

We were thinking of the updraft intensity, and the presence of hail and intense precipitation. We wanted to emphasize that a link between cloud dynamics, microphysics and thunderstorm electrification has already been demonstrated. We changed the sentence to:

*It has also been shown that a fast increase in the lightning activity, i.e. "lightning jump", is related to thunderstorm intensification, in terms of updraft intensity and the presence of hail and intense precipitation.*

Line 44: Please add a reference for MTG LI.

The following reference has been added line 42:

Kokou, P., Willemsen, P., Lekouara, M., Arioua, M., Mora, A., Van den Braembussche, P., Neri, E., and Aminou, D. M. A.: Algorithmic Chain for Lightning Detection and False Event Filtering Based on the MTG Lightning Imager, IEEE Transactions on Geoscience and Remote Sensing, 56, 5115–5124, https://doi.org/10.1109/TGRS.2018.2808965, 2018.

Footnote 1 (p. 2): An expanded discussion of the authors' reflection on this topic would be valuable. Is the reason for the change sufficient to risk the confusion that could result from two, actively-used names for the same product?

There is already some confusion around this quantity in the literature, whether on its denomination or its unit: Murphy and Demetriades (2005) called it the "flash extent density" in flashes $km^{-2}$, for Mansell (2014) it is the "flash-extent density" in counts $min^{-1}$, it was called the "accumulated flash density" in flashes $min^{-1}$ $km^{-2}$ by Bovalo et al. (2016), etc.

Our objective was to designate this quantity properly and to use the adequate unit. The term "density" was misleading because it would refer here to a count of points over a surface unit. If two adjacent pixels are merged, the number of points are added up. But satellite-measured lightning flashes are not points since the size of a flash is often larger than a pixel. If a flash illuminates two adjacent pixels, the number of flashes measured would still be 1 if the pixels are merged. In other words, this quantity cannot be integrated as a density should be able to be. In consequence, the term "density" is not suited to describe this quantity. We then recommend the term "flash extent accumulation" (FEA), measured in flashes.

The footnote was rephrase as:

*Note that the FEA has been referred to as flash extent density (FED) in former studies but is often introduced with different units (flashes $km^{-2}$, counts $min^{-1}$,...). For the sake of clarification, the terminology FEA was adopted here, expressed in flashes.*

Line 104: are the vertical winds not among the prognostic variables?

The vertical wind is not a prognostic variable of AROME. According to Seity et al. (2011), Section 2, AROME uses twelve 3D prognostic variables: 2 components of the horizontal wind (U and V), temperature T, specific content of water vapor $q_v$, rain $q_r$, snow $q_s$, graupel $q_g$, cloud droplets $q_c$, ice crystals $q_i$, turbulent kinetic energy TKE, and two nonhydrostatic variables, $\hat{q}$ and d, the vertical

divergence, that are related to pressure and vertical momentum, and are described in Eq. (1) and (2) of Seity et al. (2011). The vertical velocity at each level is retrieved using the surface vertical velocity and the vertical divergence.

Line 150: Is the limitation in maximum vertical velocity due to model resolution/numerics? Model integration typically starts to act as a low-pass filter at about 6 times the grid spacing. If it is not due to this, what is the cause?

In our simulations, the vertical velocities in convective regions are roughly between 3 and 15 m/s in general but they can reach values higher than 40 m/s: see *x* axis in panel (b) of Figure 8. However, those values occur in regions that have a very limited horizontal extension and that is why this proxy is not adapted (See answer to question #8 of the first Reviewer for more details). There can be an underestimation of vertical velocities in the models due to the fact that they are only partially resolved but it remains marginal.

Line 214: "others" should be "other"

It has been modified, thank you.

Line 235: here, proxy refers to the NWP parameterizations, correct? The FEA grid is also a proxy for real MTG LI measurements, so it might help to state which proxy is meant.

The term "proxy" refers to the 8 variables listed in section 2.3 (ice water path, graupel mass, updraft volume etc.). We specify at line 66 that those variables will be referred to as "the proxies" throughout the article. They are calculated using the AROME-France variables, listed in Section 2.2 hence they have a horizontal resolution of 1.3 km whereas the FEA has a horizontal resolution of 7 km so they need to be projected on the same grid. This sentence has been modified in the revised manuscript at line 238 as such:

*First, the proxies are calculated with the 1 h AROME-France forecasts at a horizontal resolution of 1.3 km. Then, they are projected on the FEA 7 km-grid by selecting the closest value to the FEA pixel center.*

Line 240: When discarding values equal to zero, is any rounding or other rule applied to determine what counts as zero?

No, it is only the values exactly equal to zero that have been removed. A lot of values were strictly equal to zero for some proxies, for example the updraft volume, and the objective here was mainly to reduce the dataset by discarding data that do not carry information, to limit the computation time. It corresponds to the step #5 described next question.

Somewhere between lines 240 and 245, I don't quite follow how the stacked, ranked ordered data (the histograms in fig. 4?) are used to perform the regressions. On line 250, is the value of "proxy" a normalized count at some flash rate, and not the flash rate itself? I'm probably missing something

obvious, but could the authors illustrate / visualize the regression for the simple, linear case? What goes on the x axis and y axis of the regression? How many samples are there?

We make the assumption that the proxy is an increasing function of the FEA which means that the sorted values of FEA will be compared against the sorted values of proxy to fit a regression function.

Here are the data processing steps illustrated with some diagrams (the data values and placement are randomly chosen):

Step #1: after the re-projection of the proxies values on the FEA grid, we have a grid of FEA and a grid of proxy values of the same dimensions (175 rows and 174 columns) for each hour. Example below for a grid of 4x3.

FEA

| 0 | 0 | 0 |
|---|---|---|
| 0 | 2 | 1 |
| 0 | 5 | 0 |
| 0 | 1 | 0 |

proxy

| 0 | 0 | 9 |
|---|---|---|
| 2 | 4 | 6 |
| 6 | 1 | 2 |
| 3 | 1 | 7 |

Step #2: The grids are flattened.

FEA | proxy
---|---

| FEA | proxy |
|---|---|
| 0 | 0 |
| 0 | 0 |
| 0 | 9 |
| 0 | 2 |
| 2 | 4 |
| 1 | 6 |
| 0 | 6 |
| 5 | 1 |
| 0 | 2 |
| 0 | 3 |
| 1 | 1 |
| 0 | 7 |

Step #3: the flattened grids from the other hours and other days are added up (not shown in the example). So we have 174 x 175 x 24 x 44 ~ $10^7$ samples. This number of samples has been added to the manuscript at line 240.

Step #4: the FEA and the proxy are sorted.

| FEA | proxy |
|-----|-------|
| 0 | 0 |
| 0 | 0 |
| 0 | 1 |
| 0 | 1 |
| 0 | 2 |
| 0 | 2 |
| 0 | 3 |
| 0 | 4 |
| 1 | 6 |
| 1 | 6 |
| 2 | 7 |
| 5 | 9 |

Step #5: All lines (in the FEA column and proxy column) with proxy and FEA equal to zero are removed, because they do not carry any information. In our example, it means the first two lines of the FEA and the proxy are removed.

Step #6: FEA and proxy are both normalized to the [0;1] range (not shown in the example). If there are too much orders of magnitude between the x and y ranges, some ML regression models cannot converge.

For the regression, we wish to find FEA = $f$(proxy) so the FEA is on the $y$ axis and the proxy is on the $x$ axis: the (FEA; proxy) couples are plotted as white dots in each subplots of Figure 6. One can notice these dots follow a monotonic increasing curve since the data are sorted. Since we have a huge number of samples ($10^7$) we wanted to represent the distribution of those points: the color shades in the background of Figure 6. It shows a predominance of low values.

Because the data from all the hours of all the days are regrouped and sorted, one can say they no longer have time and space dependence.

Line 261: I thought the authors removed spatial considerations from their method (line 238), so how is FSS calculated?

The relationship FEA=$f$(proxy) is indeed established with data without time and space dependence. But to validate this relationship we compare (using the FSS) an observed FEA grid to the one obtained by applying $f$ to a proxy grid. Applying $f$ to the proxy grid yields one FEA value for each proxy value while keeping the original proxy grid shape. The following sentence was added at line 268:

*The validation consists in i) verifying if the established regression models fit a dataset independent to the one it was trained with and ii) comparing the resemblance between a FEA field calculated with the regression models and the observed FEA, using the fraction skill score (FSS).*

Line 400: the combination does seem to do somewhat better over Corsica, a region highlighted by the authors as performing poorly on line 313.

Indeed, thank you for pointing that out. It seems to result from the contribution of wmax. It will be cited as an example line 393 as such:

*The general areal coverage is very similar to what was predicted with the graupel mass alone [...] with some additional isolated patches that can be attributed to the contribution of wmax, for example over Corsica.*

Fig. 14: It's somewhat interesting that a multivariate proxy makes the 1 fl/min performance worse, but the 10 fl/min performance is slightly improved. I agree with the authors that it's hard to see a benefit over simply using graupel mass.

We agree.

Line 434: the decision to not accumulate graupel mass may seem strange to some readers, but I think it makes sense if the goal is data assimilation. In DA systems, observations are often assimilated against the model state at a single time step, and so the goal here is to find a representative accumulation window for use in DA. Is this in fact the authors' motivation for the design of the experiment in this section?

Yes, exactly. For now, the data assimilation time period in AROME-France in 1 hour, meaning that observations measured ±30 minutes before and after the analysis time are assimilated, at most. In most 3DVar-like data assimilation systems, the model variables are only available at the analysis time so it is not possible to accumulate them. This information has been added to the manuscript at lines 443-5.

We wanted to know up to what accumulation period the FEAs modelled with a non-accumulated proxy were realistic.

Lines 461-3: This sentence is missing the results of the bootstrap test.

The sentence is "*the confidence intervals […] overlap*".

Line 465-6: High flash rates are not frequent in these data, so the authors' conclusion here is, in general, justified if the goal is data assimilation for regional convective structure. However, there are some modeling systems (e.g, the US NSSL's Warn-on-Forecast system) that have the aim of correctly assimilating the state of individual storm cells, where high flash rate fluctuations that capture storm state on short time scales are of more importance. Fig. 19a diverges at high flash rates, indicating that the authors' conclusion might not apply if the goal is high-flash-rate single cells. In that case, would the recommendation be to use a shorter accumulation interval, consistent with the observation that the curves converge at shorter accumulation intervals?

What this sensitivity study highlights is that the FER-proxy relationship (which proxy value corresponds to which FER value) does not depend much on the accumulation period of the FER: when assimilated, a FER data value, whether accumulated for 5 or 60 minutes, will correct the model background toward a very similar value (that is what Fig. 19a shows), except for very high FER values. In the case of those very high FER values, according to the regression functions plotted in Fig. 19a, the assimilation of a FER value accumulated for a long period will shift the model background toward a higher proxy value than if accumulated for a shorter period. To check which FER-proxy relationship is the closest to the reality, we compared observed FER to the ones obtained with the regression functions for each accumulation period. Below are the observed FER accumulated over various period around 17 UTC 26 May 2018 (Fig. 18 in the article) and the FER modelled from the regression functions for the same accumulation periods (not shown in the article):

[Figure]

The differences between the modelled FER for the different accumulation periods are almost impossible to distinguish (right panel). Even so, the fraction skill score with the 1 fl/min threshold (FER "highest values", yellow, orange and red colors in the above Figures), as plotted in Figure 20 right panel, is lower when the FER are accumulated for 60 minutes. It means that the higher the FER values are and the higher the accumulation period is, the more displacement error we get. We did not try longer accumulation periods since the AROME-France data assimilation period is 1 hour but this displacement error is expected to grow with the accumulation period. So we recommend an accumulation period shorter than 60 minutes, whether for high FER values or in general. Also, even though it was not shown in the article, an accumulation period shorter than 5 minutes will probably not be enough to gather enough lightning data to have a proper description of the thundercloud's extension.

Those recommendations were added in the revised version of the manuscript at the end of the conclusion, lines 526-8.

Line 482: I was briefly confused that the authors were recommending a multi-parameter estimation method and were not using the single-variable graupel mass operator that their analysis to this point

seemed to prefer. Please clarify that *either* IWP or F2 is preferred, because that single proxy variable is calculated from more than one underlying model state variable.

Indeed, the recommendation might be confusing. We definitely recommend a microphysical proxy, so either the graupel mass, the IWP, the rimed particle column or F2. We do not recommend a combination of microphysical and dynamical proxies because the dynamical proxies do not seem to add pertinent information when modeling the FEA spatial distribution and amplitude.

If the goal is to use the FEA=*f*(proxy) relationship to provide a lightning diagnostic, any of the microphysical proxies is recommended.

In a context of variational data assimilation, the assimilation would be impossible in the case of a completely cloud-free background because the gradient of the microphysical-related observation operator would be zero. Some work is in progress to overcome this issue but an observation operator based on several microphysical variables could increase the sensitivity to the observations since it increases the chances to have a non-zero microphysical content in the background. That is why we recommend the use of the IWP or F2 *in a data assimilation context*. This clarification has been added to the manuscript's conclusion.

**References used in the answers:**

Bovalo, C., C. Barthe et J. Pinty, 2019: Examining relationships between cloud-resolving model parameters and total flash rates to generate lightning density maps. Quarterly Journal of the Royal Meteorological Society. https://doi.org/10.1002/qj.3494

Erdmann, F., O. Caumont, and E. Defer, 2022: A geostationary lightning pseudo-observation generator utilizing low frequency ground-based lightning observations. Journal of Atmospheric and Oceanic Technology, 391, 3 –30. https://doi.org/10.1175/JTECH-D-20-0160.1

Erdmann, F., 2020: Préparation à l'utilisation des observations de l'imageur d'éclairs de Météosat Troisième Génération pour la prévision numérique à courte échéance (Preparation for the use of Meteosat Third Generation Lightning Imager observations in short-term numerical weather prediction). PhD thesis. Toulouse, France: Université Toulouse III – Paul Sabatier, http://thesesups.ups-tlse.fr/4947/

Mansell, E. R., 2014: Storm-scale ensemble Kalman filter assimilation of total lightning flash-extent data. Monthly Weather Review, 142(10), 3683-3695. https://doi.org/10.1175/MWR-D-14-00061.1

Murphy, M. J. and N. W. S. Demetriades, 2005: An analysis of lightning holes in a DFW supercell storm using total lightning and radar information. In: First Conf. on Meteorological Applications of Lightning Data. 2.3. American Meteorological Society. https://ams.confex.com/ams/Annual2005/techprogram/paper_84505.htm

Price, C. and Rind, D., 1992: A simple lightning parameterization for calculating global lightning distributions, Journal of Geophysical Research, 97, 9919–9933, https://doi.org/10.1029/92JD00719

Seity, Y., Brousseau, P., Malardel, S., Hello, G., Bénard, P., Bouttier, F., Lac, C., and Masson, V., 2011. The AROME-France Convective-Scale Operational Model, *Monthly Weather Review*, *139*(3), 976-991. https://doi.org/10.1175/2010MWR3425.1